# Model Merging Scaling Laws in Large Language Models

Yuanyi Wang [* 1]   Yanggan Gu [* 1]   Yiming Zhang [* 1]   Qi Zhou [1]   Zhaoyi Yan [2]
Congkai Xie [2]   Xinyao Wang [3 †]   Jianbo Yuan [3 †]   Hongxia Yang [1 2 4 ‡]

## Abstract

We study empirical scaling laws for language model merging measured by cross-entropy. Despite its wide practical use, merging lacks a quantitative rule that predicts returns as we add experts or scale the model size. We identify a compact power law that links model size and expert number: the size-dependent floor decreases with model capacity, while the merging tail exhibits clear diminishing returns in the number of experts. The law holds in-domain and cross-domain, tightly fits measured curves across diverse architectures and methods (Average, TA, TIES, DARE), and explains two robust regularities: most gains arrive early, and variability shrinks as more experts are included. Building on this, we present a simple theory that explains why gains fall roughly as $1/k$ and links the floor and tail to properties of the base model and the diversity across domains. This law enables *predictive planning*: estimating how many experts are needed to reach a target loss, deciding when to stop adding experts, and trading off scaling the base model versus adding experts under a fixed budget. These results make merging a predictable, budget-aware alternative to multitask fine-tuning. Our code and models are available at https://github.com/InfiXAI/Merging-Scaling-Law

## 1. Introduction

Large language models (LLMs) are often specialized by fine-tuning on different domains, producing multiple domain

---
[*]Equal contribution

[‡]Corresponding author.

[†]Work done outside of Amazon.

[1]The Hong Kong Polytechnic University (PolyU)

[2]InfiX.ai  [3]Amazon  [4]PolyU-Daya Bay Technology and Innovation Research Institute.   Correspondence to:   <yuanyi.wang@connect.polyu.hk,   hongxia.yang@polyu.edu.hk>.

*Proceedings of the $43^{rd}$ International Conference on Machine Learning*, Seoul, South Korea. PMLR 306, 2026. Copyright 2026 by the author(s).

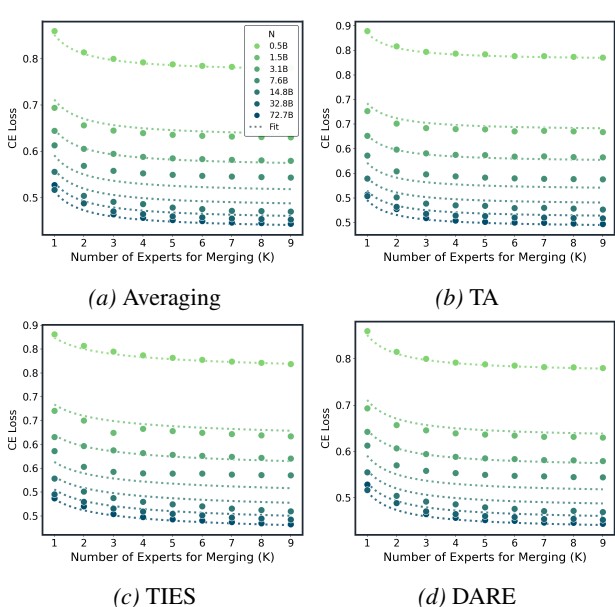

*Figure 1.* Model Merging Scaling Law. CE vs. number of merged experts ($k$) at multiple model sizes ($N$) for four merging methods. Dots are real measurements; dotted lines are fits to the unified law.

experts. Model merging combines these experts in weight space to synthesize a single model without retraining. This idea underlies a range of methods: linear rules such as weight averaging (Izmailov et al., 2018; Wortsman et al., 2022), task arithmetic (Ilharco et al.), selective or nonlinear schemes like TIES (Yadav et al., 2023), and DARE (Yu et al., 2024). Merging has proven attractive in practice—it can approximate joint training at a fraction of the cost, supports modular pipelines with adapters, e.g., LoRA (Hu et al., 2022; Mao et al., 2025; Zhou et al., 2026), and enables composition under privacy or compute constraints (Shi et al., 2026; Zhou et al., 2025).

Despite this promise, merging remains largely empirical. Practitioners experiment with subsets, orders, and normalization rules, often at substantial computational expense. Unlike pretraining, where well-established *scaling laws* guide how loss decreases with model size, data, or compute (Kaplan et al., 2020; Hoffmann et al., 2022), merging lacks an analogous quantitative account. This gap makes it difficult to anticipate convergence as more experts are added,

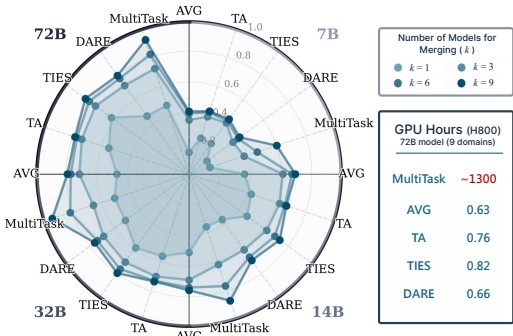

*Figure 2.* Overview of merging vs. multitask SFT. The polar axis represents the normalized negative loss.

to compare rules across base sizes, or to make budget-aware design choices.

In this paper, we first introduce a compact, predictive *merging scaling law* that couples model size $N$ with the number of merged experts $k$:

$$\mathbb{E}[L \mid N, k] \;=\; \underbrace{L_* + B\,N^{-\beta}}_{\text{floor } L_\infty(N)} \;+\; \underbrace{\frac{A_0\,N^{-\gamma}}{k + b}}_{\text{merging tail}}, \quad (1)$$

where $\beta, \gamma > 0$, $b \geq 0$. Intuitively, larger base models depress the size-dependent floor $L_\infty(N)$ and shrink the tail amplitude $A_0 N^{-\gamma}$; adding experts yields steep early improvements that taper as $1/(k+b)$. The term $L_*$ denotes the irreducible floor that remains even for very large $N$.

As shown in Fig. 1 and Fig. 2, our experiments across 10,866 merged models, base sizes from 0.5B to 72B, nine domains, and four methods (Average, Task Arithmetic (TA), TIES, and DARE) validate this power law and directly compare *merging* with *multitask SFT* under normalized loss and GPU-hours. Empirically, merging approaches multitask SFT performance while using negligible GPU-hours, and method gaps compress as $k$ and $N$ grow. Across methods, we see the same pattern: steep early gains that flatten into a $1/(k+b)$ tail, and a uniform downward shift with larger $N$ (both the floor and the tail shrink). Method differences become smaller as scale increases. $R^2 > 0.98$ over all fitted points. These findings position merging as a practical, budget-aware alternative to comprehensive multitask training and highlight the proposed merging scaling law as a tool for forecasting returns and planning budgets.

This study reveals a consistent power law for LLM merging that aligns with the later sections: (i) *larger models are easier to merge*, floors decrease with $N$ and tails shrink (Fig. 4); (ii) *most gains arrive early*, with a clear elbow at small $k$ (Section 3.3.3); (iii) *mixing domains helps pooled generalization* under the same floor+tail scaling (Section 3.3.2); (iv) *method differences are small at scale*, with both means

and variability converging (Section 3.3.4); (v) *order sensitivity fades quickly* as $k$ grows (Section 4.3); and (vi) *the power law transfers across backbones* with the same shape (Section 4.4).

In summary, this work provides:
(1) **Unified scaling law:** We introduce a compact floor+tail law that links base size and expert count, and show it applies consistently in both in-domain and cross-domain settings.
(2) **Large-scale validation:** Across extensive experiments covering diverse domains, model sizes, 10,866 models, and merging methods, the law tightly fits measured curves, variance contracts with more experts, and method gaps compress as scale increases.
(3) **Theory:** We derive a leading-order inverse-$k$ tail and variance under equal-normalized composition of effective updates, and clarify how this average-case result should be interpreted for practical preprocessing rules such as TIES and DARE.
(4) **Operational recipe:** We introduce a lightweight three-point fitting procedure that predicts the full merging curve and identifies an efficient expert count, enabling budget-aware planning. The procedure is robust to candidate-pool size and transfers across architectures.

## 2. Background, Related Work, and Setup

**Notation.** Let $N$ denote the size of the base model, $\mathcal{M}$ denotes a set of expert models, and let $k$ be the number of expert models to be merged. We denote the base model by $\theta_0$. A task vector $v$ is defined as the parameter difference between the base model and a domain-adapted model, which may be either the full parameter difference or a low-rank adaptation such as an adapter or LoRA module (Hu et al., 2022) restricted to its subspace. Unless otherwise stated, we employ *equal-weight merging*, where all task vectors are assigned the same importance. For fixed $N$ and $k$, the *expected loss* refers to the average performance over all possible $k$-element subsets of experts drawn from $\mathcal{M}$, while *variance* measures the variability of the loss.

### 2.1. Background

**Model Merging:** Model merging is the integration of multiple independently trained models into a single cohesive model by aggregating their parameters (Matena & Raffel, 2022; Jin et al., 2022; Wang et al., 2025a). Existing work performs merging either (i) on the *full parameter space*, like model soups and Fisher weight-space averaging (Izmailov et al., 2018; Wortsman et al., 2022; Davari & Belilovsky, 2024), or (ii) within *modular subspaces*, most commonly adapters or LoRA (Hu et al., 2022), enabling plug-and-play composition across domains with minimal interference (Hu et al., 2022; Mao et al., 2025). Merging methods are refined with advanced techniques (Jhunjhunwala et al.,

2024; Yan et al., 2025; Akiba et al., 2025), including dynamic parameter selection (Yang et al., 2023). Despite these advances, the core idea remains manipulating *task vectors*—changes relative to the base pre-trained model (Rinaldi et al., 2025; Zhang et al., 2024; Bowen et al., 2024). Further gains come from processing task vectors before aggregation, for instance using element-wise masks or gates (e.g., TIES/DARE) to reduce conflicts between experts (Yadav et al., 2023; Yu et al., 2024; Lu et al., 2024; Wang et al., 2026). These methods cover the majority of practical pipelines and constitute the settings evaluated in this paper. However, most of aforementioned studies consider limited expert models to merge, and the relation between the number of experts and the effectiveness is underexplored. (Wang et al., 2025c; Yadav et al., 2024) examined this relationship from theoretical and empirical perspectives, respectively, identifying factors that influence merging performance, but did not provide a systematic scaling law to guide merging across different domains and model sizes.

**Scaling Law:** Classical scaling laws quantify how loss scales with *model size*, *data*, and *compute*: parameter/data power laws and compute-optimal trade-offs (Kaplan et al., 2020; Hoffmann et al., 2022; Hestness et al., 2017). Extensions study transfer and evaluation efficiency, as well as precision/quantization scaling that augments the usual size–data laws with a precision term (Kumar et al.). Scaling laws provide a predictable, quantitative framework that helps researchers make more informed decisions and prevent the blind allocation of vast resources (Ardalani et al., 2022; Klug & Heckel, 2022; Neumann & Gros, 2022; Geiping et al., 2022). Specifically, scaling laws have been leveraged by (Filipovich et al., 2022) to empirically demonstrate that Direct Feedback Alignment (DFA) is not a more compute-efficient training method than backpropagation. (Hilton et al., 2023) extend these laws by incorporating sparsity, finding a compute-optimal sparse-dense trade-off that challenges the conventional belief that dense models are always superior for large-scale training. (Fernandes et al., 2023) research on scaling laws to multilingual neural machine translation models, revealing that data mixture weights affect the multiplicative factor of the scaling law but not the scaling exponent. These laws guide pretraining, but they do not address *composition in weight space*.

### 2.2. Setup

**Expert Models:** We use a dual–track design to balance control and realism (details in Appendix D). *(i) Controlled experts:* Starting from the same base, we train nine domain experts with identical hyperparameters. All base models are from the Qwen2.5 series (0.5B–72B) (Qwen et al., 2025). *(ii) Open-source experts:* We additionally treat diverse HuggingFace checkpoints as experts to test robustness under heterogeneous, partly opaque post-training.

**Data:** We construct our own expert set $\mathcal{M}$ using data from Mixture-of-Thoughts (Face, 2025) and Open-Science[1], where all solutions are generated by DeepSeek-R1 (DeepSeek-AI et al., 2025) to ensure consistent quality. For mathematics, we sample 93,700 instances and categorize them into five subfields (Algebra, Analysis, Discrete Mathematics and Combinatorics, Geometry and Topology, Number Theory), with 200 medium-difficulty problems per subfield reserved for validation. For science, we combine both datasets, selecting 20,000 training and 200 validation samples from each of Biology, Physics, and Chemistry. For code, we use 82,000 training and 10,000 validation samples from Mixture-of-Thoughts. This construction provides broad domain coverage, balanced validation sets, and consistent standards across all expert models.

**Merging $k$ Experts:** In this paper, we study four merging methods: Average merge, TA, TIES, and DARE. Table 1 gives a unified form for these recipes. For a given number of experts $k$, we denote by $\mathcal{K} = \{K \subseteq \mathcal{M} : |K| = k\}$ the collection of all $k$-expert subsets of $\mathcal{M}$. Merging all experts can be written as:

$$\theta = \theta_0 + \sum_{i \in K} \alpha_{i,k} \Psi(v_i), \qquad \sum_{i \in K} \alpha_{i,k} = c \quad (2)$$

with a fixed scale $c > 0$ (often $c = 1$). Here $\Psi$ is the rule-specific preprocessing map. For Average and TA, $\Psi(v) = v$; for TIES and DARE, $\Psi$ includes trimming, masking, sparsification, or rescaling before the equal-normalized composition. Thus these practical rules can be viewed as composing transformed *effective updates* rather than introducing external information at merge time.

**Expert capacity:** We treat base size $N$ and expert count $k$ as the explicit scaling axes and keep the expert-training recipe fixed in the controlled Qwen experiments. Expert capacity is therefore not modeled as a separate axis; it enters through the distribution of effective updates. Changing the LoRA rank, adapter width, fine-tuning token budget, or expert quality would alter the mean direction, covariance, and curvature alignment of $\Psi(v_i)$, thereby shifting the fitted floor $L_\infty(N)$, tail amplitude $A(N)$, and possibly their exponents. Modeling expert capacity as a third scaling axis is a natural extension of the present two-axis law.

**Evaluation:** We report token-level cross-entropy: per domain, we score 30M held-out tokens and average the loss. For each $k$, we aggregate by averaging CE over all $\binom{|\mathcal{M}|}{k}$ expert subsets (or a uniform random subset when $N > 8$B to control cost; details are provided in Appendix E).

---

[1]https://huggingface.co/datasets/nvidia/OpenScience

# 3. Scaling Laws with Merging Experts and Model Size

In this section, we ask a simple question: *As we merge more experts (k) and use larger models (N), how does the cross-entropy (CE) loss change?* We study this in two standard setups: *in-domain* (evaluation on the single domain) and *cross-domain* (experts drawn from nine heterogeneous domains and evaluated by macro-averaging over all nine). We use four widely adopted merge rules that scale from small to large models: Average (Wortsman et al., 2022), TA (Ilharco et al.), TIES (Yadav et al., 2023), and DARE (Yu et al., 2024). Our grids cover $N \in \{0.5, 1.5, 3, 7, 14, 32, 72\}$B (with 10,866 models in total) and $k \in \{1, \ldots, 9\}$; domains are *algebra, analysis, geometry, discrete, number_theory, code, chemistry, physics, biology*.

**Construction of the expected loss.** For each backbone size $N$, we start from a single base checkpoint and train $M=9$ domain-specialist experts. Given a merge rule and a target expert number $k$, there are $\binom{M}{k}$ possible expert subsets. For each $(N, k)$, we merge either all subsets (when feasible) or a large uniform sample, and evaluate the cross-entropy loss $L(N, k, s)$ of the merged model on held-out data, where $s$ indexes the subset.

We define the *expected merge loss* at $(N, k)$ as the empirical average over subsets,

$$\widehat{\mathbb{E}}[L \mid N, k] \; = \; \frac{1}{S_{N,k}} \sum_{s=1}^{S_{N,k}} L(N, k, s),$$

where $S_{N,k}$ denotes the number of sampled subsets.[2]

The first two panels of Fig. 3 visualize this construction on representative Qwen-2.5 models. These points correspond to losses from different expert subsets rather than a density over data samples; any apparent two-band structure reflects heterogeneity across subsets, while our analysis focuses on the subset-averaged expectation. While individual subset losses exhibit nontrivial variability, the per-$k$ mean $\widehat{\mathbb{E}}[L \mid N, k]$ forms a smooth, monotonic curve with diminishing returns as $k$ increases. This motivates modeling the *expected* behavior rather than individual expert combinations. Additional results are provided in Appendix G.

## 3.1. A Unified Empirical Scaling Law

Let $\mathcal{M}$ denote the set of $M$ experts for a given backbone size $N$, and let $K \subseteq \mathcal{M}$ be a subset of size $k$. For a fixed $(N, k)$, choosing $K$ uniformly at random among all $\binom{M}{k}$ subsets and applying a merge rule yields a random merged loss $L$. Throughout this subsection, we therefore study the

---

[2]In our grids, $S_{N,k}$ equals the full $\binom{M}{k}$ whenever feasible; otherwise we use a large uniform sample, which yields visually indistinguishable curves.

*conditional expectation* $\mathbb{E}[L \mid N, k]$ over the random choice of $K$.

Empirically, we find that this expected loss admits a simple and interpretable *floor + tail* form with a small finite-$k$ offset:

$$\mathbb{E}[L \mid N, k] = L_\infty(N) \; + \; \frac{A(N)}{k + b}, \;\; b \geq 0 \text{ (small)}. \tag{3}$$

Here $L_\infty(N)$ is the limiting "best models can do" as $k \to \infty$, and $A(N)/(k+b)$ is a diminishing-returns term that explains why most gains arrive by small $k$. Both size dependencies are well captured by simple power laws:

$$L_\infty(N) = L_* + B \, N^{-\beta}, \; A(N) = A_0 \, N^{-\gamma}, \; \beta, \gamma \geq 0. \tag{4}$$

*Interpretation.* Bigger models help twice: they lower the floor $L_\infty(N)$ and shrink the tail amplitude $A(N)$, so (i) CE is lower for any fixed $k$, and (ii) fewer experts are needed to get close to the floor.

To fit this power law, we estimate $(L_*, B, \beta, A_0, \gamma, b)$ with weighted nonlinear least squares. Because the empirical variability across runs contracts roughly like $1/k$, we use weights proportional to $k$ when fitting curves in $k$ (this stabilizes early-$k$ noise without over-fitting the tail). All methods and both setups yield near-unity $R^2$ with small, structureless residuals; a tiny $b$ absorbs occasional early-$k$ curvature. Fig. 1 plots CE vs. the number of merged experts $k$ at multiple model sizes $N$ for each method; dots are measurements and dotted lines are the fitted $L_\infty(N) + A(N)/(k+b)$ curves. The same visual pattern holds across methods: steep early gains that flatten into a $1/(k+b)$ tail, and a uniform downward shift as $N$ increases.

### 3.1.1. IN-DOMAIN

Fig. 3 shows the Average merging performance in the single algebra domain, and all domains are provided in Appendix H.0.1. We can observe that: **(1) Diminishing returns in $k$.** Within each domain, CE decreases monotonically (or near-monotonically) as we merge more experts and follows the $1/(k+b)$ tail predicted by equation 3. Most of the achievable improvement arrives early: there is a clear elbow by $k \approx 5 \sim 6$, after which additional experts yield progressively smaller gains. **(2) Scaling with $N$.** Bigger models help in two orthogonal ways consistent with equation 4: the floor $L_\infty(N)$ drops with $N$ and the tail amplitude $A(N)$ is flat-to-decreasing, so (i) CE is lower at any fixed $k$, and (ii) fewer experts are needed to approach the floor. Math-like domains exhibit shorter tails (earlier saturation), whereas science-like domains benefit more from increasing $k$ before saturating.

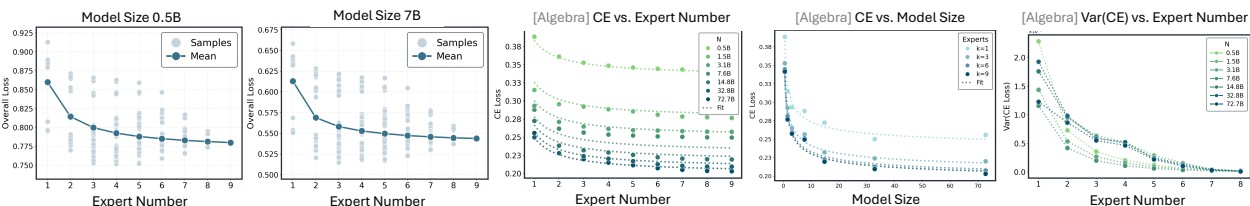

*Figure 3.* Empirical construction and in-domain scaling example. Panels (1)–(2) show $\mathbb{E}[L \mid N, k]$ on Qwen-2.5 models at fixed sizes ($N$=0.5B, 7B): light points are individual expert subsets and the solid curve is the empirical mean at each $k$. Panels (3)–(5) show the single-domain algebra case: CE vs. number of merged experts $k$, CE vs. model size $N$, and the subset-level CE variance as $k$ increases. Dots are measurements; lines are fits to $L_\infty(N) + A(N)/(k + b)$.

### 3.1.2. CROSS-DOMAIN

Fig. 1 shows the cross-domain power law across nine domains as the expert count varies, and panels (3)–(5) of Fig. 3 show the corresponding model-size fit and variance trend in a representative in-domain setting. We observe two patterns: (1) Same law, pooled over domains. When merging experts drawn across heterogeneous domains and evaluating by macro-averaged CE, the same floor+tail law equation 3 holds: gains are monotone with $k$, steep early, and flatten into a $1/(k+b)$ tail. The elbow again occurs around $k \approx 5$. (2) Scaling with $N$. Increasing model size uniformly shifts curves downward (lower floor) and weakly contracts tails (smaller $A(N)$), mirroring the in-domain behavior: larger models are both better at any fixed $k$ and require fewer experts to approach the floor.

Across both *in-domain* and *cross-domain* settings, the expected merge loss fits the same power law (Equation equation 3). Bigger $N$ lowers the floor and shortens the tail, explaining the monotone gains and early saturation in $k$.

### 3.2. Theory for the Merging Scaling Law

This section explains why the *average-case* performance of merging $k$ experts exhibits a leading-order $1/k$ tail, and how this behavior couples with model size $N$ to yield the joint scaling law used in our fits. Under equal normalization, merging corresponds to averaging task update vectors. As $k$ increases, the variance of the averaged update shrinks as $1/k$, and a Taylor second-order expansion of the loss converts this variance reduction into an expected-loss improvement of the same order. This mechanism depends only on first- and second-order statistics in the merged subspace and is agnostic to task semantics. For practical preprocessing rules, we apply the argument to the effective update $\tilde{v}_i = \Psi(v_i)$: TIES and DARE change the mean and covariance by trimming, masking, sparsifying, or rescaling updates before composition, but the equal-normalized aggregation still has the same leading variance scaling when these effective updates have stable second moments.

**Setup and Assumptions.** Fix a model size $N$. Let $L(\cdot; N)$

be twice continuously differentiable near the base $\theta_0(N)$ with $M(N)$-Lipschitz Hessian $H(N)$ and gradient $g(N)$. Expert/task update vectors $v(N)$ lie in the merged subspace with mean $\mu(N)$, covariance $\Sigma(N)$, and finite sixth moment. For rules with preprocessing, we interpret $v(N)$ below as the effective update $\tilde{v}(N) = \Psi(v(N))$ after the rule-specific transformation. We use *equal-normalization* $\alpha_{i,k} = c/k$ (covering uniform averaging, normalized sums, adapter ensembling, and the normalized composition step of TIES/DARE after preprocessing); specialized non-uniform or learned weightings can change the tail rate and are outside the scope of this theorem.

Under these assumptions, we can derive a precise asymptotic characterization of the population-averaged loss as a function of the number of merged experts $k$.

**Theorem 3.1** (Average-case joint merging law). *Under the assumptions above (equal weights), for each fixed $N$ the population-averaged loss over $k$ merged experts satisfies the second-order law*

$$\boxed{\mathbb{E}[L \mid N, k] = L_\infty(N) + \frac{A(N)}{k} + \mathcal{O}_N(k^{-3/2})}$$

$$\text{with} \quad L_\infty(N) = L(\theta_0; N) + c\,g^\top\mu + \tfrac{1}{2}\,c^2\,\mu^\top H\,\mu,$$
$$A(N) = \tfrac{1}{2}\,c^2\,\mathrm{Tr}(H\,\Sigma). \tag{5}$$

*where $H$ denotes an approximation to the Hessian matrix, and $\mu, \Sigma$ represent respectively the mean and covariance of task vectors in the merged subspace. In particular, the empirical family equation 3 appears with $b(N) = 0$ at leading order; finite-$k$ effects manifest as a small positive offset in practice. Parameterizing $L_\infty(N), A(N)$ by equation 4 yields the practical joint model $\mathbb{E}[L \mid N, k] = L_* + BN^{-\beta} + A_0 N^{-\gamma}/(k + b_0)$.*

**Proof:** The proof is provided in Appendix B.

Theorem 3.1 separates the merging behavior into two components: an asymptotic performance limit $L_\infty(N)$ and a finite-$k$ improvement term $A(N)/k$. The former captures the loss attained as $k \to \infty$, determined by the base model, the mean task direction, and local curvature, while the latter

governs the rate at which this limit is approached through the curvature, covariance interaction $\text{Tr}(H\Sigma)$. Crucially, the $1/k$ decay is universal under equal normalization of the effective updates, with all remaining effects strictly lower order.

From an empirical perspective, this result directly motivates the functional form of our merging scaling law. The observed $k$-dependence follows from the theorem at leading order, while the additional offset $b_0$ accounts for finite-$k$ effects and curvature-surrogate mismatches. This yields a simple yet expressive joint scaling model, which we validate experimentally across architectures and domains.

**Scope for TIES and DARE.** The theorem provides a leading-order statement about equal-normalized composition of effective updates; it is not a rule-specific derivation of every trimming, election, masking, sparsification, or rescaling step. These preprocessing steps change $\mu(N)$, $\Sigma(N)$, and their alignment with local curvature; empirically, these changes are absorbed into the fitted floor $L_\infty(N)$, tail amplitude $A(N)$, offset $b$, or small bounded finite-$k$ deviations. Under this interpretation, the same floor+tail law fits Average, TA, TIES, and DARE with high $R^2$.

Beyond the mean trend, the same analysis also characterizes the stability of merging, showing that variability across different subsets of experts decreases as $k$ grows.

**Corollary 3.2.** *Let $a_N \triangleq g(N) + H(N)\, c\, \mu(N)$. Under the same assumptions and $a_N^\top \Sigma(N)\, a_N > 0$,*

$$\text{Var}\big(L(\theta_0 + \Delta\theta_k; N)\big) = \Theta\Big(\frac{1}{k}\Big), \ \ \text{sd} = \mathcal{O}\Big(\frac{1}{\sqrt{k}}\Big).$$

*If $a_N^\top \Sigma(N)\, a_N = 0$,*

*the variance contracts faster, at $\Theta(1/k^2)$.*

**Proof:** The proof is provided in Appendix C.

Corollary 3.2 shows that merging more experts improves not only accuracy but also reliability. In the generic case, the standard deviation of the loss decays as $1/\sqrt{k}$, indicating increasing concentration around the mean scaling curve. This variance shrinkage explains the empirical observation that large-$k$ merges exhibit both better average performance and reduced run-to-run variability (Appendices H and I).

### 3.3. Core Findings for Merging

#### 3.3.1. LARGER MODELS MAKE MERGING EASIER

**Setup:** We study the *in-domain* case across 9 domains and define "easier to merge" as: at a fixed number of experts $k$, (i) CE is lower, and (ii) the number of experts needed to get $\varepsilon$-close to the domain floor is smaller. Following the law in Section 3, we estimate the *floor $L_\infty(N)$* and the *tail amplitude $A(N)$* from joint $(N, k)$ fits and summarize them in Fig. 4.

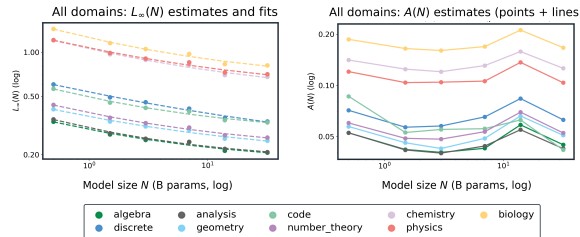

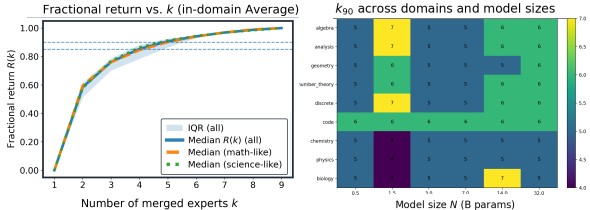

*(a)* Per-domain floors $L_\infty(N)$ and tail amplitudes $A(N)$ as functions of model size.

*(b)* Fractional return $R(k)$ and the smallest $k$ that reaches a target 90% return.

*Figure 4.* Larger models are easier to merge, and most gains arrive early.

**Findings.** The floor curves in Fig. 4a decay cleanly with model size $N$ across all domains (power-law trend), while the tail-amplitude curves in the same panel are small and overall flat-to-decreasing as $N$ grows. Together these two effects explain why larger models are easier to merge: at any fixed $k$ the CE is lower and fewer experts are required to approach the floor. As a headline number, at $k=9$ the domain-averaged CE drops from $0.739$ (@0.5B) to $0.430$ (@32B), a 41.9% reduction. Domains with shorter tails (math-like) saturate earlier; science-like domains benefit more from increasing $k$ but still follow the same floor+tail pattern. The fractional-return summary in Fig. 4b shows that $k = 5$ and $k = 6$ cross the $85\%/90\%$ thresholds, respectively. Thus, roughly 60% of the nine-expert pool is enough to recover over 90% of the measured improvement. Per-domain parameters and worked examples for the experts-to-floor budget are provided in Appendix J.1.

#### 3.3.2. MIXING DOMAINS HELPS GENERALIZATION

**Findings & why.** As seen in Fig. 1 and Fig. 4a, cross-domain merging follows the *same* law as in-domain: gains are monotone in $k$, steep early, and flatten into a $1/(k+b)$ tail, with an elbow around $k\approx5$. Larger $N$ uniformly shifts the pooled curves downward, mirroring the lower floor and smaller tail amplitude from Section 3.3.1. The diversity of donors reduces domain-specific bias (lower $L_\infty$) while averaging attenuates variance and leaves a short tail governed by $A(N)/(k+b)$. The small bounded non-monotonicity observed in the fitted tail coefficient as $N$ varies does not propagate to the overall loss, confirming that cross-domain

generalization *inherits* the same diminishing-returns scaling.

### 3.3.3. GAINS CONCENTRATE IN EARLY EXPERTS

**Setup.** We quantify the return from merging $k$ experts at a fixed $(N, d)$ by the fraction of realized improvement $R(N, d, k)$ computed from the monotone envelope of the measured CE curve (see Appendix J.2). We summarize the median $R(k)$ over all $(N, d)$ with an IQR band, together with the $k_{90}$ heatmap, in Fig. 4b.

**Findings & why:** As shown in Fig. 4b, most of the improvement arrives early: the median curve crosses $85\%$ by $k=5$ and $90\%$ by $k=6$, and the $k_{90}$ heatmap concentrates in $\{5, 6\}$ across domains and model sizes. Math-like domains tend to saturate slightly earlier, while science-like domains keep a longer—but still flattening—tail. This "early elbow" follows directly from the unified law $L(N, k) = L_\infty(N) + A(N)/(k+b)$: the marginal gain $\Delta_k \approx A(N)/[(k+b)(k+1+b)]$ decays roughly as $k^{-2}$, so returns diminish sharply beyond the first few experts.

### 3.3.4. METHODS DIFFER LITTLE AT LARGE SCALE

**Setup.** We compare four merge methods, Average, TA ($\lambda=0.8$), TIES ($\lambda\in\{0.5, 1\}$), and DARE (density 0.2), under the same protocol as before, reporting macro-averaged CE across nine domains and fitting each curve with the unified law. Fig. 5a shows mean CE vs. $k$ at $N=32$B; Fig. 5b shows the corresponding merge-to-merge variance.

**Findings & why:** As $k$ grows (and especially at larger $N$), method gaps in *mean* CE compress quickly: in Fig. 5a, small early advantages (TA/TIES at $k\leq3$) shrink to a tight band by $k\approx8$ (differences $\lesssim 2\%$). Variance exhibits the same convergence (Fig. 5b), contracting near $\sim 1/k$ and approaching a small floor where all methods meet. This behavior follows directly from the shared scaling form: the diminishing-returns tail $A(N)/(k+b)$ makes early steps method-sensitive, while the common floor $L_\infty(N)$ dominates at larger $k$ and $N$, leaving only second-order differences. The results are consistent with the observations of (Yadav et al., 2024), further confirming their findings.

## 4. Further Analysis and Recipe

Beyond establishing the unified law in Section 3, we stress–test it along practical axes that affect day-to-day merging: how large the candidate pool is, whether mixing domains helps, how sensitive results are to order/selection, and whether findings transfer across backbones. Throughout, we keep the evaluation protocol fixed and re-estimate the same $L_\infty(N) + A(N)/(k+b)$ family. The main text reports trends and takeaways; per-domain numbers and fit diagnostics are in the Appendix.

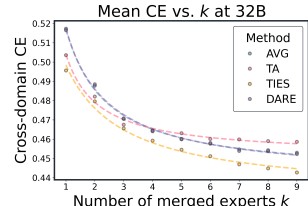 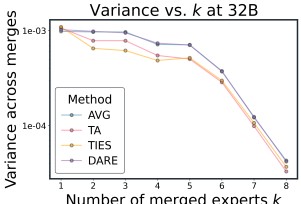

*(a)* Mean CE vs. $k$ at $N=32$B. *(b)* Merge-to-merge variance vs. $k$ at $N=32$B.

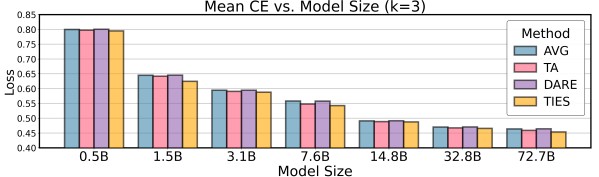

*(c)* Mean CE vs. model size at $k=3$.

*Figure 5.* Method sensitivity diminishes at scale. Mean CE and variance follow a common power law across methods; small early-$k$ gaps narrow quickly, and variance shows near-$1/k$ contraction for all methods.

### 4.1. Does a bigger candidate pool help?

**Setup.** We repeat the cross-domain analysis while *restricting* the pool of available donor domains to $M \in \{8, 7\}$ (DARE; identical $(N, k)$ grids), then refit the unified law. Fig. 6 contrasts the fitted floor $L_\infty(N)$ and tail $A(N)$ for $M=8$ vs. $M=7$.

**Findings & why:** The law itself is *stable* to pool size: floors remain tight power laws in $N$ with negligible change across $M$ (Figs. 6a and 6b). The effect of a larger pool shows up almost entirely in the *tail*: moving from $M=8$ to $M=7$ makes $A(N)$ *flat-to-decreasing* with $N$ on science-like domains (chemistry/physics) while leaving math-like domains nearly unchanged. Intuitively, a slightly more diverse pool supplies complementary donors and reduces residual cross-domain mismatch, shrinking the $A(N)/(k+b)$ term; this yields the clearest gains at moderate-to-large $k$ and larger $N$. In short, a bigger pool chiefly helps by tightening the tail rather than shifting the floor.

### 4.2. Can three points predict the whole $k$-curve? (Yes)

**Setup.** For each series, either a single *(domain, N)* in-domain curve or a *(method, N)* cross-domain curve, we fit the unified law $L(k) = L_\infty(N) + \frac{A(N)}{k+b}$ using only the first three points $k \in \{1, 2, 4\}$, then forecast the full $k\in\{1, \ldots, 9\}$ trajectory and the value at a target $k$.

**Findings & why:** Three points suffice. Across domains and methods, the early-$k$ slope plus the long-tail shape are captured well by $L_\infty+A/(k+b)$, so fitting on $\{1, 2, 4\}$ closely tracks the full curve in Fig. 7a. The implied $k^\star$ concentrates around $5\sim6$ in Fig. 7b, aligning with the elbow found in

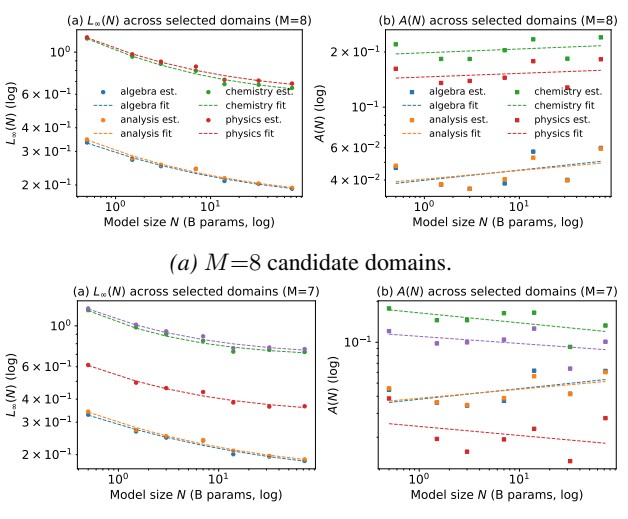

*(a) $M{=}8$ candidate domains.*

*(b) $M{=}7$ candidate domains.*

*Figure 6.* Unified-law fits with reduced candidate pools ($M{=}8, 7$). Across domains, floors $L_\infty(N)$ remain stable, while tail terms $A(N)$ exhibit weak or no shrinkage with $N$.

Section 3.3.3. Intuitively, the model's floor $L_\infty$ anchors the late regime while $A$ controls the early drop; those two degrees of freedom are identifiable from three well-spaced points, yielding stable forecasts without overfitting. Thus, early measurements are sufficient for budget-aware merge planning.

### 4.3. Does merge order matter?

**Setup.** We permute donor orders under DARE, and at each $(N, k)$ summarize the across-order dispersion of the macro-averaged CE by standard deviation, range, and coefficient of variation; we also fit a parsimonious tail $\text{Std}_{\text{order}}(N, k) \approx c_0(N) + c_1(N)/(k+b)$.

**Findings.** Order effects fade fast. Fig. 8a shows that, at 32B, both the interquartile mass and the whiskers collapse as $k$ grows (about $83\%$ shrinkage in whisker length by $k{=}8$). Fig. 8b confirms that this contraction holds *for all* base sizes and follows the same $1/(k+b)$ pattern that governs the mean, with larger $N$ being slightly more stable. Fig. 8c quantifies worst–best differences: the relative range reduction at $k{=}9$ is $\approx 24\%$ (0.5B), $32\%$ (32B), and $34\%$ (72B), and in absolute CE the 32B spread falls from $\sim 0.086$ ($@k{=}1$) to $\sim 0.015$ ($@k{=}8$). Practically, once $k \geq 6$ the across-order spread is small compared to early-$k$ method gaps and to the floor itself, so curating a specific merge order yields little benefit.

### 4.4. Does the same law hold on other backbones?

**Setup.** We replicate the power-law from Section 3 on two open-source backbones, LLaMA-3.2-3B, LLaMA-3-

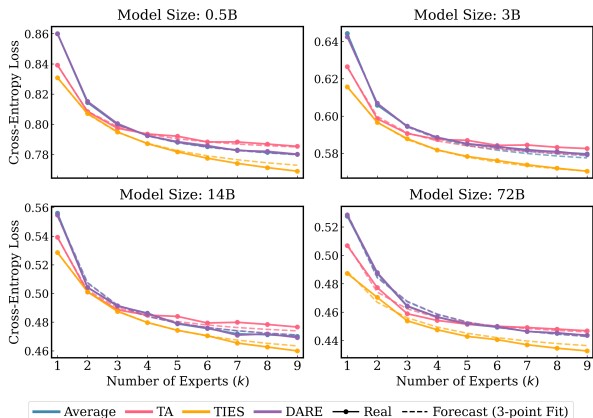

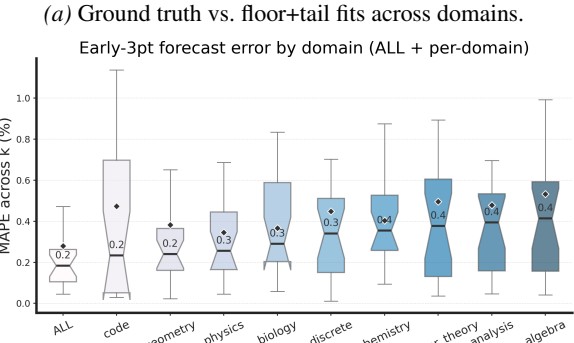

*(a) Ground truth vs. floor+tail fits across domains.*

*(b) Forecast error across $k$ and the induced distribution of recommended $k^\star$.*

*Figure 7.* Predicting the $k$-curve from three points. Forecast errors stay low and the recommended $k^\star$ concentrates at small values.

8B, and Gemma2 family. For each backbone, we merge experts sampled across nine domains, report the macro-averaged CE for $k \in \{1, \ldots, 9\}$, and fit the same floor+tail form $L(k) = L_\infty + \frac{A}{k+b}$ with a small $b$. To complement the main *curve fit*, we also visualize the *marginal gain* $\Delta L(k) = L(k{-}1) - L(k)$ and the *experts-to-target* bars $k^\star_{80/90}$ (the smallest $k$ that reaches 80/90% of the total $k{=}1 \rightarrow 9$ improvement).

**Findings.** Both backbones follow the same inverse-tail law: macro CE decreases monotonically in $k$, with steep early gains that flatten thereafter (Fig. 9a). The floor+tail model fits the data extremely well ($R^2{=}0.999$ for LLaMA-3.2 3B and $R^2{=}0.995$ for LLaMA-3 8B, details listed in Appendix L.1), confirming quantitative consistency across backbones. The marginal gain curves (Fig. 9b) decay smoothly with $k$, and both models reach roughly 80% of the total improvement with only $k \approx 4{-}5$ experts. Absolute CE levels differ modestly, LLaMA-3.2 3B sits lower with a slightly steeper early slope, reflecting backbone capability rather than a change of scaling law. Additional results on Gemma 2 also follow the same form, shown in Appendix K

**Note 1:** We also report downstream evaluations in Ap-

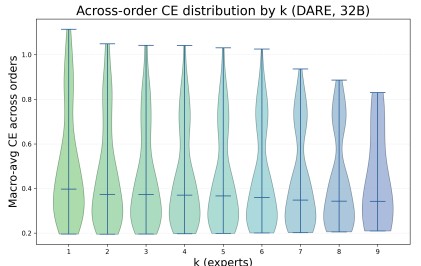
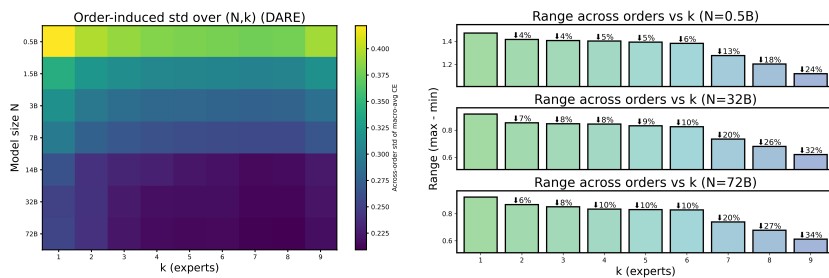

*(a)* CE distribution across orders at $N=32$B.

*(b)* Across-order standard deviation over $N$ *(c)* Worst–best range at representative sizes. and $k$.

*Figure 8.* Order sensitivity contracts with $k$ (DARE). The distribution, standard deviation, and worst–best range all tighten rapidly as $k$ increases.

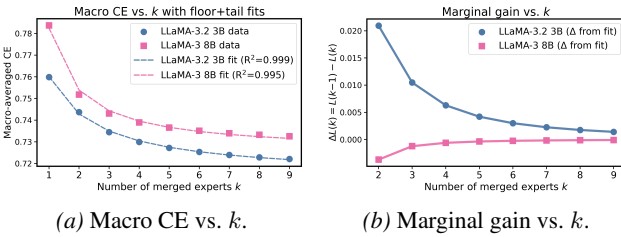

*(a)* Macro CE vs. $k$.   *(b)* Marginal gain vs. $k$.

*Figure 9.* Cross-backbone validation on LLaMA. Macro CE follows the same inverse tail on LLaMA-3.2 3B and LLaMA-3 8B, and marginal gains decay smoothly with $k$.

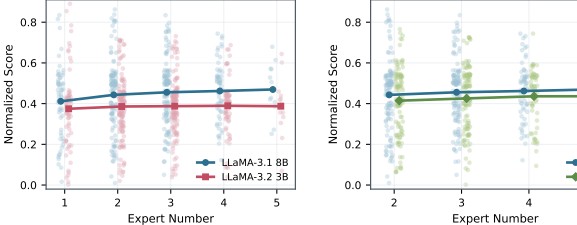

*(a)* TA across LLaMA 3.1-8B & 3.2-3B backbones.   *(b)* TA vs. TIES on 8B.

*Figure 10.* Downstream scores exhibit early gains and saturation. Translucent points denote normalized benchmark scores from individual expert subsets; filled markers connected by solid lines denote empirical means at fixed $k$.

pendix K. Aggregated task metrics generally improve with $k$ and then plateau, showing the same diminishing-return pattern as CE at a qualitative level. However, downstream scores can saturate earlier than token-level CE, and we do not claim that they follow the same law quantitatively.

**Note 2:** We further extend the cross-domain experiments to a 16-domain pool on the LLaMA-3B backbone (original 9 domains plus Japanese, medical, house-arrangement, Korean, emotion, elementary school mathematics, and Java-code experts), and the aggregated cross-entropy still follows the same power law (see Appendix L).

To make the downstream trend explicit in the main paper, Fig. 10 summarises representative results from Appendix K. Under Task Arithmetic, both LLaMA backbones improve quickly from $k=1$ to $k\approx3$ and then flatten, with the smaller 3B model saturating earlier. On the 8B backbone, TA and TIES follow the same qualitative trajectory for $k\geq2$: most utility is captured by the first few experts, and merge-rule differences are concentrated in the early-$k$ regime. The translucent point clouds also show that task-level metrics are substantially noisier than token-level CE, even when their filled mean markers follow a stable trend. Thus downstream evaluation supports the same qualitative diminishing-return picture, while making clear that benchmark scores can plateau earlier and should not be read as following the exact CE scaling law.

## 5. Conclusion

This paper presented a simple, predictive *merging scaling law* that links model size and the number of merged experts via a floor+tail power law. This law unifies a broad set of empirical regularities: larger bases lower the size-dependent floor, most improvement arrives at small $k$, variance contracts with additional experts, method gaps compress at scale, and merge order quickly becomes inconsequential. The same power law form holds in-domain and cross-domain, and transfers across architectures and representative merging methods. Beyond description, the law is prescriptive. A lightweight fit from a few early points forecasts the full loss–vs.–$k$ curve and recommends an efficient expert count, enabling budget-aware decisions about when to stop adding experts and how to trade off scaling the base model versus increasing $k$. Expert capacity, such as LoRA rank, adapter width, or fine-tuning token budget, is absorbed into the fitted floor and tail in this study and remains a natural additional scaling axis. Together, these results elevate merging from heuristic practice to a computationally efficient, budget-aware alternative to multitask fine-tuning.

## Acknowledgements

This paper is fully supported by a grant from the Research Grants Council of the Hong Kong Special Administrative Region, China (Project No. T41-517/25-N).

## Impact Statement

This work aims to advance the understanding of model merging by providing a principled scaling law that characterizes how performance evolves with model size and the number of merged experts. By offering theoretical insight and practical guidance for efficient expert merging, our results may help reduce unnecessary computation and resource usage in large-scale model development. The techniques studied here operate on trained models and do not introduce new learning objectives or data sources, and thus inherit the same ethical considerations as existing large language models. While model merging can potentially lower the cost of deploying specialized capabilities, it may also contribute to the broader accessibility of powerful models, with societal impacts that are well studied in the machine learning literature. We do not foresee any specific negative ethical consequences unique to this work beyond those already associated with large-scale machine learning systems.

## Reproducibility statement

We have made every effort to ensure that the results reported in this work are reproducible. All models and datasets employed are publicly available, and we describe the methodological choices, data sources, and evaluation protocols in detail in Section 2 of the main text. Additional implementation details and hyperparameters are documented in Appendix D. Furthermore, we provide the complete source code as supplementary material to facilitate replication and independent verification. Our checkpoints will also be released.

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

*Table 1.* Unified view of model merging recipes

| Method | $\Psi(v)$ | $c$ | $\alpha$ | Add. Params |
|---|---|---|---|---|
| Average | $v$ | 1 | $\alpha = 1/k$ | - |
| TA | $v$ | 0.8 | $\alpha = 1/k$ | - |
| TIES | Trim, Elect, Disjoint | 1 | $\alpha = 1/k$ | $d = 1.0$ |
| DARE | $m \odot v/(1-p)$ | 1 | $\alpha = 1/k$ | $p = 0.2$ |

## Statement of LLMs Usage

We utilized a Large Language Model (LLM) solely as **an editing tool** for syntactic error correction and stylistic enhancement. It is important to note that the LLM **did not** participate in any aspect of the core research, such as the generation or revision of the central research ideas, the design of experiments, or the overall organization and chapter arrangement of this paper.

## Limitations.

Our main claim concerns expected token-level cross-entropy under merging. This choice is deliberate: CE is dense, relatively low-variance, and directly aligned with the local second-order analysis that gives rise to the floor+tail form. Downstream benchmark scores provide complementary evidence of utility, but they need not obey the same scaling law. In practice, they can plateau earlier than CE as $k$ grows, since they are sparser, more thresholded, and typically noisier aggregates of task success. We therefore view the contribution of this paper as a predictive law for merge loss and a principled way to reason about the tradeoff between model size and expert count, rather than as an exact predictor of every downstream benchmark curve. Our study centers on cross-entropy and equal-normalized composition; extending to other objectives and adaptive weighting is an important next step. While the law is robust across datasets, methods, and backbones we tested, probing extreme scales, additional modalities, and broader downstream metrics (robustness, safety, calibration) remains future work. We also keep expert capacity controlled rather than treating it as a third scaling axis; changing LoRA rank, adapter width, training tokens, or expert quality should alter the effective-update statistics and therefore the fitted floor/tail parameters. On the theoretical side, refining the link between floor/tail parameters, curvature anisotropy, and domain dispersion, and developing selection/ordering policies that exploit these quantities, could further tighten predictions and automate practical merging at scale.

## A. Model Merging Recipes

We use a unified form to represent all of these recipes in Table 1.

## B. Detailed proof of Theorem 3.1

We fix a model size $N$ and omit $(N)$ when clear. Following Assumption 3.2: (i) $L$ is twice continuously differentiable near $\theta_0$ with an $M$-Lipschitz Hessian; (ii) task vectors $v_i$ are i.i.d. with mean $\mu$ and covariance $\Sigma$, and $\mathbb{E}\|v_i\|^6 < \infty$; (iii) equal-weight normalisation $\alpha_{i,k} = c/k$.

**Decomposition.** Let

$$\Delta\theta_k(S) = \sum_{i \in S} \tfrac{c}{k} v_i = c\mu + \varepsilon_k(S), \qquad \varepsilon_k(S) := \tfrac{c}{k} \sum_{i \in S} (v_i - \mu).$$

Expectation $\mathbb{E}[\cdot]$ is with respect to the uniform random $k$-subset $S$ (the same orders follow for i.i.d. sampling with replacement) and $\varepsilon$ means the sampling error.

**Lemma B.1** (Moments of the mean-corrected step). $\mathbb{E}[\varepsilon_k] = 0$ *and* $\mathbb{E}[\varepsilon_k \varepsilon_k^\top] = \tfrac{c^2}{k}\Sigma$. *Moreover,* $\mathbb{E}\|\varepsilon_k\|^3 = \mathcal{O}(k^{-3/2})$ *under* $\mathbb{E}\|v_i\|^6 < \infty$.

*Proof.* Linearity gives $\mathbb{E}[\varepsilon_k] = 0$. For the second moment, averaging $k$ i.i.d. centred vectors yields covariance $c^2\Sigma/k$. The

$p = 3$ Marcinkiewicz–Zygmund (Ortega-Cerdà & Saludes, 2007; Ibragimov & Sharakhmetov, 1999) inequality gives

$$\mathbb{E}\|\varepsilon_k\|^3 \;\leq\; \frac{C_3\,c^3}{k^{3/2}}\Big(\mathbb{E}\|\xi_1\|^2\Big)^{3/2} \;+\; \frac{C_3'\,c^3}{k^2}\mathbb{E}\|\xi_1\|^3 \;=\; \mathcal{O}\Big(\frac{1}{k^{3/2}}\Big),$$

for $\xi_i := v_i - \mu$, hence the stated rate after multiplying by $c^3$. $\qquad\square$

**Step 1: Taylor expand at $\theta_0 + c\mu$.** Define $\phi(\delta) := L(\theta_0 + c\mu + \delta)$. Let $a := \nabla\phi(0) = \nabla L(\theta_0 + c\mu)$ and $H_S := \nabla^2\phi(0) = \nabla^2 L(\theta_0 + c\mu)$. The Hessian is $M$-Lipschitz, hence the second-order Taylor formula with remainder

$$\phi(\delta) = \phi(0) + a^\top\delta + \tfrac{1}{2}\,\delta^\top H_S\delta + R_S(\delta), \qquad |R_S(\delta)| \leq \tfrac{M}{6}\|\delta\|^3. \tag{6}$$

Plugging $\delta = \varepsilon_k(S)$ and taking expectation, using Lemma B.1,

$$\begin{aligned}
\mathbb{E}\big[L(\theta_k(S))\big] &= L(\theta_0 + c\mu) + a^\top\mathbb{E}[\varepsilon_k] + \tfrac{1}{2}\,\mathbb{E}\big[\varepsilon_k^\top H_S\varepsilon_k\big] + \mathbb{E}[R_S(\varepsilon_k)] \\
&= L(\theta_0 + c\mu) + \tfrac{1}{2}\,\mathrm{Tr}\big(H_S\,\mathbb{E}[\varepsilon_k\varepsilon_k^\top]\big) + \mathbb{E}[R_S(\varepsilon_k)] \\
&= L(\theta_0 + c\mu) + \frac{1}{2}\,c^2\,\mathrm{Tr}\big(H_S\Sigma\big)\cdot\frac{1}{k} + \mathcal{O}\Big(\frac{1}{k^{3/2}}\Big).
\end{aligned} \tag{7}$$

Thus, at the *asymptote point* $\theta_0 + c\mu$ the averaged curve has a $1/k$ tail with coefficient $\tfrac{1}{2}c^2\mathrm{Tr}(H_S\Sigma)$, up to $\mathcal{O}(k^{-3/2})$.

**Step 2: relate $(L_\infty(N), A(N))$ used in the main text to the above.** In the main text we present the $k\to\infty$ intercept and tail amplitude at the *base* $\theta_0$, using a PSD curvature surrogate $H$ (e.g., GGN/Fisher) evaluated at $\theta_0$:

$$L_\infty(N) \;:=\; L(\theta_0) + c\,g^\top\mu + \tfrac{1}{2}c^2\,\mu^\top H\,\mu, \qquad A(N) \;:=\; \tfrac{1}{2}c^2\,\mathrm{Tr}(H\Sigma),$$

where $g = \nabla L(\theta_0)$.

To connect these to equation 7, apply Taylor at $\theta_0$ with the same Lipschitz-$M$ control:

$$L(\theta_0 + c\mu) = L(\theta_0) + c\,g^\top\mu + \tfrac{1}{2}c^2\,\mu^\top H\,\mu \;+\; \rho_0, \qquad |\rho_0| \;\leq\; \frac{M}{6}c^3\|\mu\|^3 \;+\; \underbrace{\tfrac{1}{2}c^2\,\big|\mu^\top(\nabla^2 L(\theta_0) - H)\mu\big|}_{\text{curvature surrogate error}}. \tag{8}$$

Similarly, since $\|H_S - \nabla^2 L(\theta_0)\|_{\mathrm{op}} \leq M\,c\|\mu\|$ (Hessian Lipschitz along the segment),

$$\mathrm{Tr}(H_S\Sigma) = \mathrm{Tr}(H\Sigma) \;+\; \eta_0, \qquad |\eta_0| \;\leq\; \|(H_S - H)\|_{\mathrm{op}}\,\mathrm{Tr}(\Sigma) \;\leq\; M\,c\|\mu\|\,\mathrm{Tr}(\Sigma) \;+\; \big|\mathrm{Tr}\big((\nabla^2 L(\theta_0) - H)\Sigma\big)\big|. \tag{9}$$

Combining equation 7–equation 9,

$$\mathbb{E}\big[L(\theta_k(S))\big] = \underbrace{L(\theta_0) + c\,g^\top\mu + \tfrac{1}{2}c^2\,\mu^\top H\,\mu}_{L_\infty(N)} \;+\; \underbrace{\tfrac{1}{2}c^2\,\mathrm{Tr}(H\Sigma)}_{A(N)}\cdot\frac{1}{k} \;+\; R_{N,k}, \tag{10}$$

with an explicit error bound

$$|R_{N,k}| \;\leq\; \underbrace{|\rho_0|}_{\mathcal{O}(\|\mu\|^3)\,+\,\text{surrogate}} \;+\; \underbrace{\frac{1}{2}c^2\,\frac{|\eta_0|}{k}}_{\mathcal{O}(\|\mu\|/k)\,+\,\text{surrogate}} \;+\; \underbrace{C\,k^{-3/2}}_{\text{from }\mathbb{E}[R_S(\varepsilon_k)]}, \tag{11}$$

where $C$ depends on $M$, $c$ and the sixth-moment bound of $v_i$. Hence,

$$\boxed{\mathbb{E}[L \mid N, k] = L_\infty(N) \;+\; \frac{A(N)}{k} \;+\; \mathcal{O}_N\Big(\frac{1}{k^{3/2}}\Big) \;+\; \mathcal{O}_N(\|\mu\|^3) \;+\; \mathcal{O}_N\Big(\frac{\|\mu\|}{k}\Big) \;+\; (\text{error}).}$$

**Interpretation of the approximation terms.** The $\mathcal{O}(k^{-3/2})$ term is the genuine *averaging* remainder from Step 1. The $\mathcal{O}(\|\mu\|^3)$ and $\mathcal{O}(\|\mu\|/k)$ terms arise from *using base-point coefficients* $(g, H)$ to parameterise the intercept and tail: when $\|\mu\|$ is moderate (typical in practice for adapter/LoRA merging or small $c$), these terms are dominated by the leading $1/k$ tail. Any persistent curvature-surrogate mismatch at $\theta_0$ is absorbed into the (fitted) $L_\infty(N)$ and $A(N)$ in the empirical model; it does not change the $1/k$ rate.

**Conclusion (Theorem 3.1 in $\approx$ form).** Collecting the above, for each fixed $N$,

$$\mathbb{E}[L \mid N, k] \approx L_\infty(N) + \frac{A(N)}{k},$$

with a quantitative remainder given by equation 11. Equivalently, at the granularity used in the main text,

$$\mathbb{E}[L \mid N, k] = L_\infty(N) + \frac{A(N)}{k} + \mathcal{O}_N\Big(\frac{1}{k^{3/2}}\Big),$$

where the $N$-dependent constants (including the small base-point/curvature-surrogate discrepancies) are absorbed into $L_\infty(N), A(N)$—exactly the form fitted in our 2D scaling law. $\qquad\square$

## C. Detailed proof of Corollary 3.2

We continue with the setting and notation of Appendix B. Fix a model size $N$ and omit $(N)$ when clear. Based on the equation 6, the second-order expansion at $\theta_0 + c\mu$:

$$L(\theta_0 + c\mu + \delta) = L(\theta_0 + c\mu) + a^\top \delta + \tfrac{1}{2}\delta^\top H_S \delta + R_S(\delta), \qquad |R_S(\delta)| \le \tfrac{M}{6}\|\delta\|^3, \tag{12}$$

with $a := \nabla L(\theta_0 + c\mu)$ and $H_S := \nabla^2 L(\theta_0 + c\mu)$. Besides Lemma B.1 (which gave $\mathbb{E}[\varepsilon_k] = 0$, $\mathrm{Cov}(\varepsilon_k) = \frac{c^2}{k}\Sigma$, and $\mathbb{E}\|\varepsilon_k\|^3 = \mathcal{O}(k^{-3/2})$), we will need $p{=}4, 6$ moment bounds. By Marcinkiewicz–Zygmund / Rosenthal inequalities (Ortega-Cerdà & Saludes, 2007),

$$\mathbb{E}\|\varepsilon_k\|^p = \mathcal{O}\big(k^{-p/2}\big), \qquad p \in \{2, 4, 6\}. \tag{13}$$

Then we make a variance decomposition. By equation 12 with $\delta = \varepsilon_k(S)$,

$$L(\theta_k(S)) = C + \underbrace{a^\top \varepsilon_k}_{L_1} + \underbrace{\tfrac{1}{2}\varepsilon_k^\top H_S \varepsilon_k}_{L_2} + \underbrace{R_S(\varepsilon_k)}_{L_3}, \qquad C := L(\theta_0 + c\mu).$$

Hence

$$\begin{aligned}
\mathrm{Var}\big(L(\theta_k(S))\big) = {} & \mathrm{Var}(L_1) + \mathrm{Var}(L_2) + \mathrm{Var}(L_3) \\
& + 2\mathrm{Cov}(L_1, L_2) + 2\mathrm{Cov}(L_1, L_3) + 2\mathrm{Cov}(L_2, L_3).
\end{aligned} \tag{14}$$

We bound the six pieces one by one.

**(i) Linear term:** $\mathrm{Var}(L_1)$. Since $\mathrm{Var}[\varepsilon_k] = 0$ and $\mathrm{Cov}(\varepsilon_k) = \frac{c^2}{k}\Sigma$,

$$\mathrm{Var}(L_1) = \mathrm{Var}(a^\top \varepsilon_k) = a^\top \mathrm{Cov}(\varepsilon_k)a = \frac{c^2}{k} a^\top \Sigma a. \tag{15}$$

**(ii) Quadratic term:** $\mathrm{Var}(L_2)$. Using $(x^\top Ax)^2 \le \|A\|_F^2 \|x\|^4$,

$$\mathbb{E}[L_2^2] = \tfrac{1}{4}\mathbb{E}\big[(\varepsilon_k^\top H_S \varepsilon_k)^2\big] \le \tfrac{1}{4}\|H_S\|_F^2 \mathbb{E}\|\varepsilon_k\|^4 = \mathcal{O}\Big(\frac{1}{k^2}\Big)$$

by equation 13 with $p{=}4$. Moreover $\mathbb{E}[L_2] = \tfrac{1}{2}\mathbb{E}[\varepsilon_k^\top H_S \varepsilon_k] = \tfrac{1}{2}\mathrm{Tr}(H_S \mathrm{Cov}(\varepsilon_k)) = \tfrac{1}{2}\frac{c^2}{k}\mathrm{Tr}(H_S \Sigma)$, so $|\mathbb{E}[L_2]| = \mathcal{O}(1/k)$, hence $|\mathbb{E}[L_2]|^2 = \mathcal{O}(1/k^2)$. Therefore

$$\mathrm{Var}(L_2) = \mathbb{E}[L_2^2] - \mathbb{E}[L_2]^2 = \mathcal{O}\Big(\frac{1}{k^2}\Big). \tag{16}$$

**(iii) Remainder:** $\mathrm{Var}(L_3)$. By equation 12, $|L_3| \le \tfrac{M}{6}\|\varepsilon_k\|^3$, so $\mathbb{E}[L_3^2] \le \big(\tfrac{M}{6}\big)^2 \mathbb{E}\|\varepsilon_k\|^6 = \mathcal{O}\big(\tfrac{1}{k^3}\big)$ by equation 13 with $p{=}6$, hence

$$\mathrm{Var}(L_3) \le \mathbb{E}[L_3^2] = \mathcal{O}\Big(\frac{1}{k^3}\Big). \tag{17}$$

**(iv) Covariances.** By Cauchy–Schwarz and the above variance bounds,

$$|\text{Cov}(L_1, L_2)| \leq \sqrt{\text{Var}(L_1)\text{Var}(L_2)} = \mathcal{O}\Big(\frac{1}{k^{3/2}}\Big), \tag{18}$$

$$|\text{Cov}(L_1, L_3)| \leq \sqrt{\text{Var}(L_1)\text{Var}(L_3)} = \mathcal{O}\Big(\frac{1}{k^2}\Big), \tag{19}$$

$$|\text{Cov}(L_2, L_3)| \leq \sqrt{\text{Var}(L_2)\text{Var}(L_3)} = \mathcal{O}\Big(\frac{1}{k^{5/2}}\Big). \tag{20}$$

Then combining equation 14–equation 20,

$$\text{Var}\big(L(\theta_k(S))\big) \;=\; \frac{c^2}{k}\, a^\top \Sigma\, a \;+\; \mathcal{O}\Big(\frac{1}{k^2}\Big). \tag{21}$$

Here $\mathcal{O}(1/k^2)$ is a *one-sided upper bound* on the remainder. If $\alpha > 0$, which is non-degenerate case, there exist constants $C_1, C_2 > 0$ and $k_0$ such that for all $k \geq k_0$,

$$\frac{C_1}{k} \;\leq\; \text{Var}\big(L(\theta_k(S))\big) \;\leq\; \frac{C_2}{k},$$

hence

$$\text{Var}\big(L(\theta_k(S))\big) \;=\; \Theta\Big(\frac{1}{k}\Big), \qquad \text{sd}\big(L(\theta_k(S))\big) \;=\; \mathcal{O}\Big(\frac{1}{\sqrt{k}}\Big).$$

For the degenerate linear term, where $a^\top \Sigma a = 0$, the linear contribution vanishes and equation 16–equation 20 give the uniform bound

$$\text{Var}\big(L(\theta_k(S))\big) = \mathcal{O}\Big(\frac{1}{k^2}\Big).$$

Moreover, whenever $H_S$ is nonzero on the range of $\Sigma$ and the fourth central moments of $v_i$ are not all degenerate along that range (a mild condition satisfied in our experiments), the quadratic fluctuation has *nonzero* variance constant, so the bound is tight:

$$\text{Var}\big(L(\theta_k(S))\big) = \Theta\Big(\frac{1}{k^2}\Big).$$

$\square$

# D. Expert Model Details

*Table 2.* Training Hyperparameters

| Hyperparameter | Value |
|---|---|
| Batch Size | 16 |
| Learning Rate | $1 \times 10^{-5}$ |
| Warmup Ratio | 0.05 |
| Number of Epochs | 2 |
| Maximum Sequence Length | 16,384 |
| Optimizer | Adam (with offloading) |
| Precision | bfloat16 |
| Gradient Checkpointing | Enabled |
| Zero Redundancy Optimizer Stage | 3 |

For evaluation, we evaluate model performance using *(token-level) cross-entropy loss*. LLM benchmark scores can vary across repeated runs and execution environments (Yang et al., 2026). Therefore, we randomly sample 30M tokens from the corresponding validation set for each domain. Let $x_t$ denote the $t$-th token sequence in the evaluation set and $p_\theta(x_t)$ the probability assigned by the model parameterized by $\theta$. The domain-specific loss is defined as the average negative log-likelihood:

$$\mathcal{L}_{\text{overall}} = -\frac{1}{\sum_{i \in \mathcal{M}} T_i} \sum_{i \in \mathcal{M}} \sum_{t=1}^{T_i} \log p_\theta(x_t | x_{t-1}, .., x_1),$$

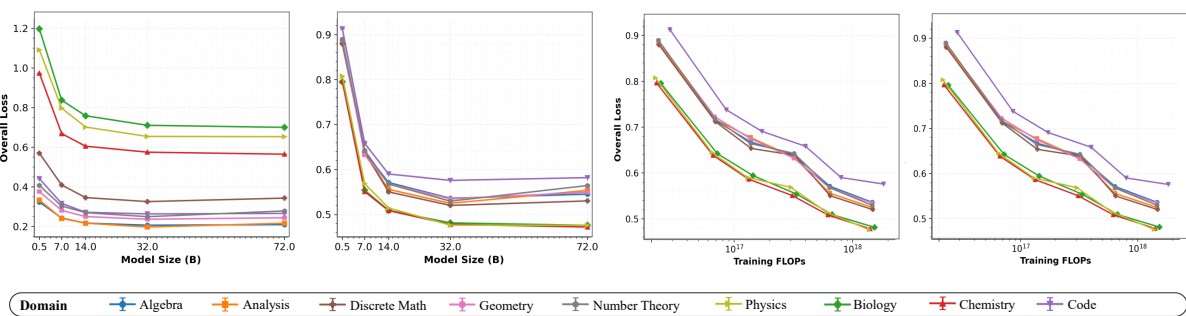

*Figure 11.* Expert Post-training Scaling Law. Expert models performance improves as we increase the model size, computational budget used for post-training.

where $T_i$ is the number of tokens in domain $i$. Even for a given $k$, we have $\binom{|\mathcal{M}|}{k}$ possible selections to merge. Each such choice yields a potentially distinct merged model. This indicates that the loss is not only a function of $k$ but also depends on which specific domains are included. Therefore, for a fixed $k$, we enumerate all $\binom{|\mathcal{M}|}{k}$ possible subsets of domain experts and compute the expected loss over them.[3]

**Note:** We isolate weight-space merging and its scaling; complementary *model fusion* baselines (e.g., InfiGFusion, InfiFPO (Wang et al., 2025b; Gu et al., 2025)) are different ways as they require data and additional training.

## E. Sampling Algorithm

---
**Algorithm 1** Diverse Permutation Generation

---
**Require:** $k \in \mathbb{N}$, base sequence $\mathbf{s} = [1, 2, \ldots, 9]$
**Ensure:** Set of $k$ diverse permutations $\mathcal{P} = \{\pi_1, \ldots, \pi_k\}$
 1: Initialize $\mathcal{P} \leftarrow \{\mathbf{s}\}$
 2: **if** $k \geq 2$ **then**
 3: $\quad$ $\mathcal{P} \leftarrow \mathcal{P} \cup \{\text{reverse}(\mathbf{s})\}$
 4: **end if**
 5: **for** $i = 3$ to $k$ **do**
 6: $\quad$ Generate candidate set $\mathcal{C}$ by random shuffling of $\mathbf{s}$ ($|\mathcal{C}| = 1000$)
 7: $\quad$ $\pi^* \leftarrow \arg\max_{\pi \in \mathcal{C}} \min_{\pi' \in \mathcal{P}} d_H(\pi, \pi')$
 8: $\quad$ $\mathcal{P} \leftarrow \mathcal{P} \cup \{\pi^*\}$
 9: **end for**
10: **Return** $\mathcal{P}$

---

We employ Algorithm 1 to perform sampling over model merge combinations, where $d_H$ denotes the Hamming distance. Fig. 12 illustrates a comparison between curves obtained via our sampling strategy (where $k = 15$) and those obtained from full merging combinations on the 0.5B model. The results demonstrate that the sampled curves closely align with the full ones, both in terms of overall trend and numerical values.

## F. Scaling Laws for Expert Model Training

In addition to investigating the scaling laws of model merging, we further examine the scaling behavior of expert models during the post-training stage. Specifically, we conduct a systematic analysis across different domains to understand how post-training affects expert models. Our study focuses on characterizing the relationship between the magnitude of the loss and three key factors: model size, the number of training tokens, and the overall computational budget. This analysis provides new insights into the scaling laws that govern post-training dynamics and highlights their potential applicability

---
[3]Since the computational overhead of model merging grows with model size, we mitigate the cost by randomly sampling a subset of all possible combinations when the model size exceeds 8B parameters. The complete sampling procedure is detailed in the next section.

across diverse domains.

Fig. 11 illustrates the performance of expert models, measured in terms of loss, as a function of model size and computational budget. Overall, we observe a consistent trend across domains: larger models and greater computation generally yield improved performance. This observation aligns with the well-established *language modeling scaling law* (Kaplan et al., 2020). Nevertheless, an important distinction arises across domains. For instance, the performance curve in the *Biology* domain exhibits substantially higher loss values compared to that in *Geometry*, even under comparable training conditions. This suggests that the model's pre-existing knowledge reserves differ across domains, leading to heterogeneous post-training dynamics despite identical training configurations. Such domain-specific disparities may further induce instability when merging expert models trained on heterogeneous knowledge bases.

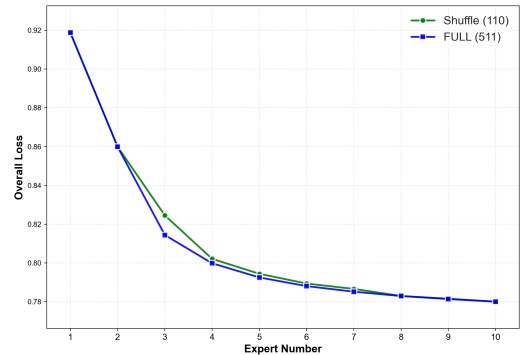

*Figure 12.* Results for different numbers of merged experts on the 0.5B model. The base model is also considered one expert.

## G. Empirical Construction of $\mathbb{E}[L \mid N, k]$

In this section, Figures illustrate the expected loss of different representative cases, where light points show individual subset losses $L(N, k, s)$ for different model sizes, $N$B, while the solid curve traces the per-$k$ mean $\widehat{\mathbb{E}}[L \mid N, k]$ that we fit our scaling law to. As $k$ grows, the scatter narrows, but the fitted curve remains smooth, which motivates modelling the *mean* behaviour rather than individual subsets.

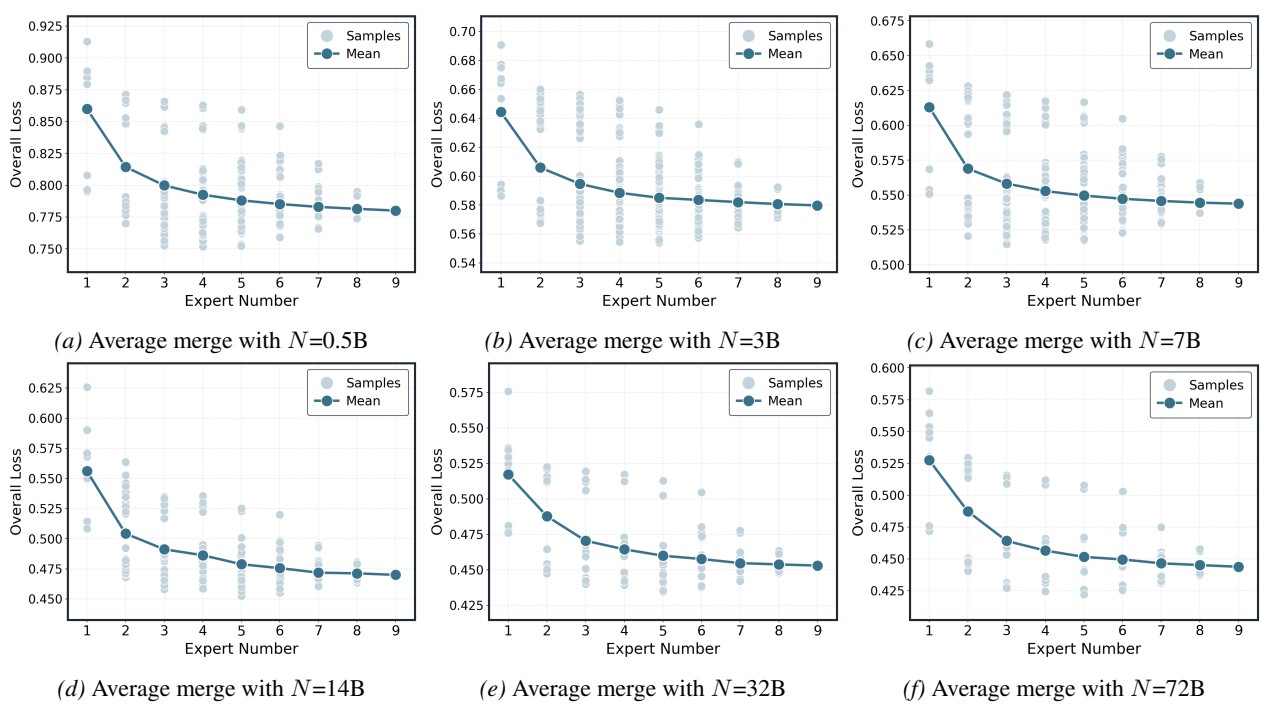

*Figure 13.* Empirical construction of $\mathbb{E}[L \mid N, k]$ in the cross-domain setting. Each figure shows average merging on Qwen 2.5 at a fixed model size. Light points are individual merged models (different expert subsets and seeds), and the solid curve is the empirical mean over all subsets at each $k$.

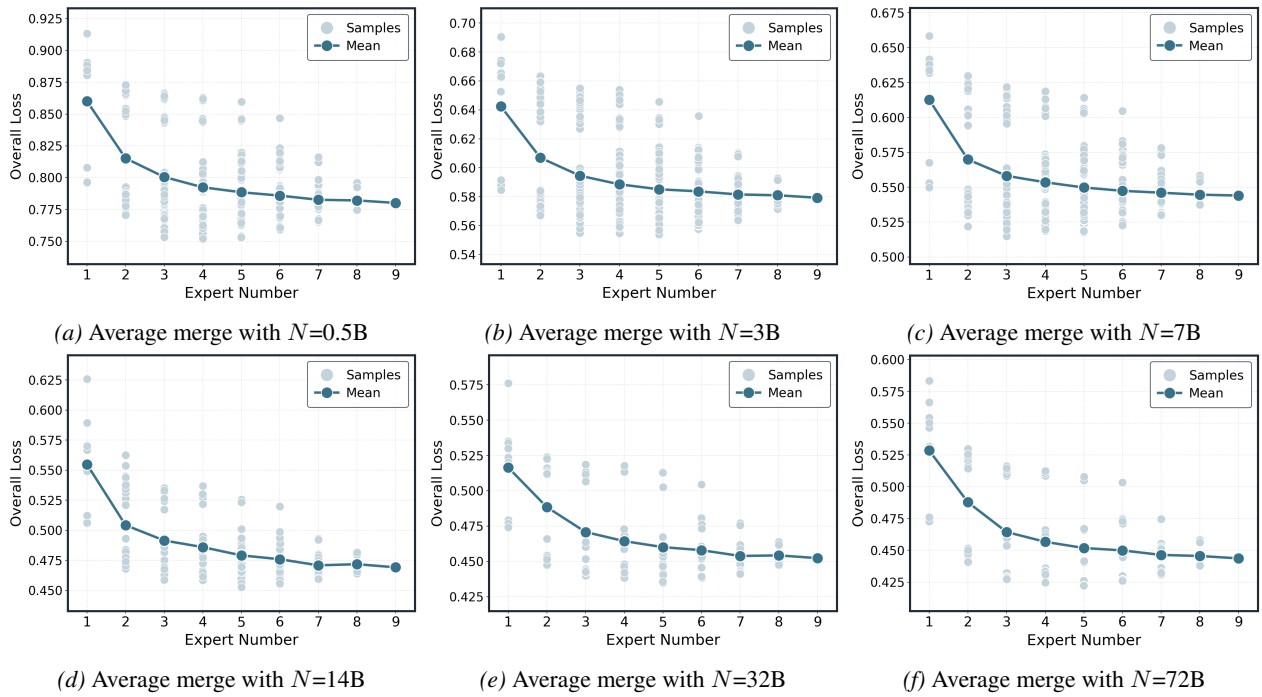

*Figure 14.* Empirical construction of $\mathbb{E}[L \mid N, k]$ in the cross-domain setting. Each figure shows DARE merging on Qwen 2.5 at a fixed model size.

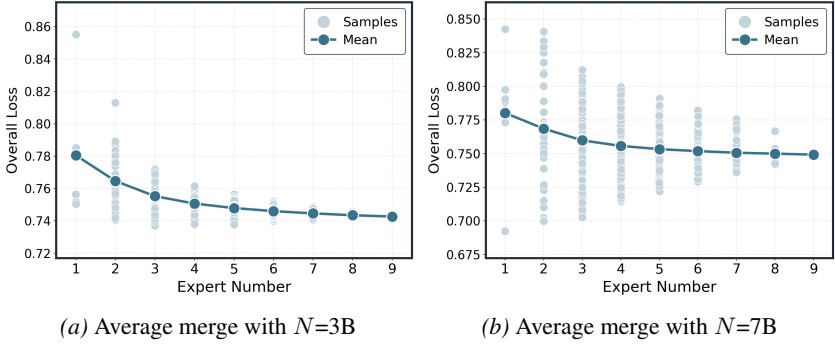

*Figure 15.* Empirical construction of $\mathbb{E}[L \mid N, k]$ in the cross-domain setting. Each figure shows TA merging on LLaMA-3.1/3.2 at a fixed model size.

# H. In-Domain Fits

### H.0.1. IN-DOMAIN (SINGLE-DOMAIN EVIDENCE)

**Diminishing returns in $k$.** CE decreases near-monotonically with $k$ and follows the $1/(k+b)$ tail. At 0.5B the macro in-domain CE drops from $\approx 0.816$ at $k=1$ to $\approx 0.739$ at $k=9$ ($-9.5\%$); at 32B it drops from $\approx 0.493$ to $\approx 0.430$ ($-12.8\%$). Most gains arrive by $k \approx 5$ (math-like domains saturate sooner; science-like domains carry longer tails).

**Scaling with $N$.** Both the floor $L_\infty(N)$ and the tail amplitude $A(N)$ shrink with $N$; at fixed $k=9$, macro CE moves from $\approx 0.739$ (@0.5B) to $\approx 0.430$ (@32B), about $-42\%$. Per-domain joint fits (Average) give tight exponents (e.g., $\beta \in [0.33, 0.42]$) and high $R^2$ (Table 3).

**Where the details live.** Full per-domain parameters for Average/TA/TIES (incl. $b$), plus 72B forecasts, are reported in Appendix H. The 72B extrapolation is modest: at $k=9$ the median in-domain CE is forecast to drop another $\sim 6$–$10\%$ from 32B to 72B.

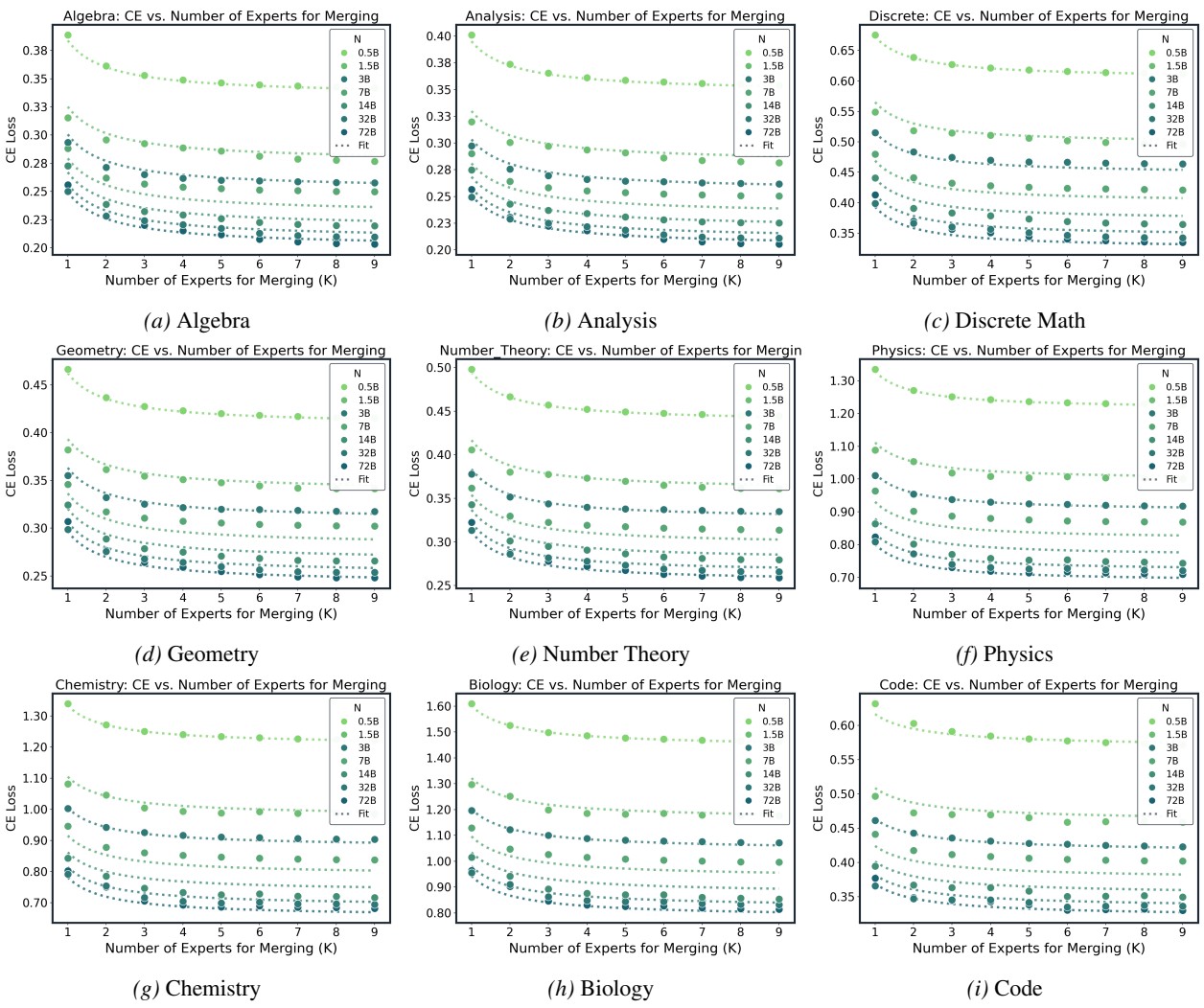

*Figure 16.* Merging Scaling Law with the Averaging Method

## H.1. Mean CE: Joint $(N, k)$ Fits

Table 3 reports the per-domain parameters of the joint law $L_{\infty,d}(N) = L_{*,d} + B_d N^{-\beta_d}$ and $A_d(N) = A_{0,d} N^{-\gamma_d}$ with the finite-$k$ offset $b_{0,d}$. All numbers come from weighted nonlinear least squares (weights $\propto k$). $R^2$ is computed on held-in $k$ grid points.

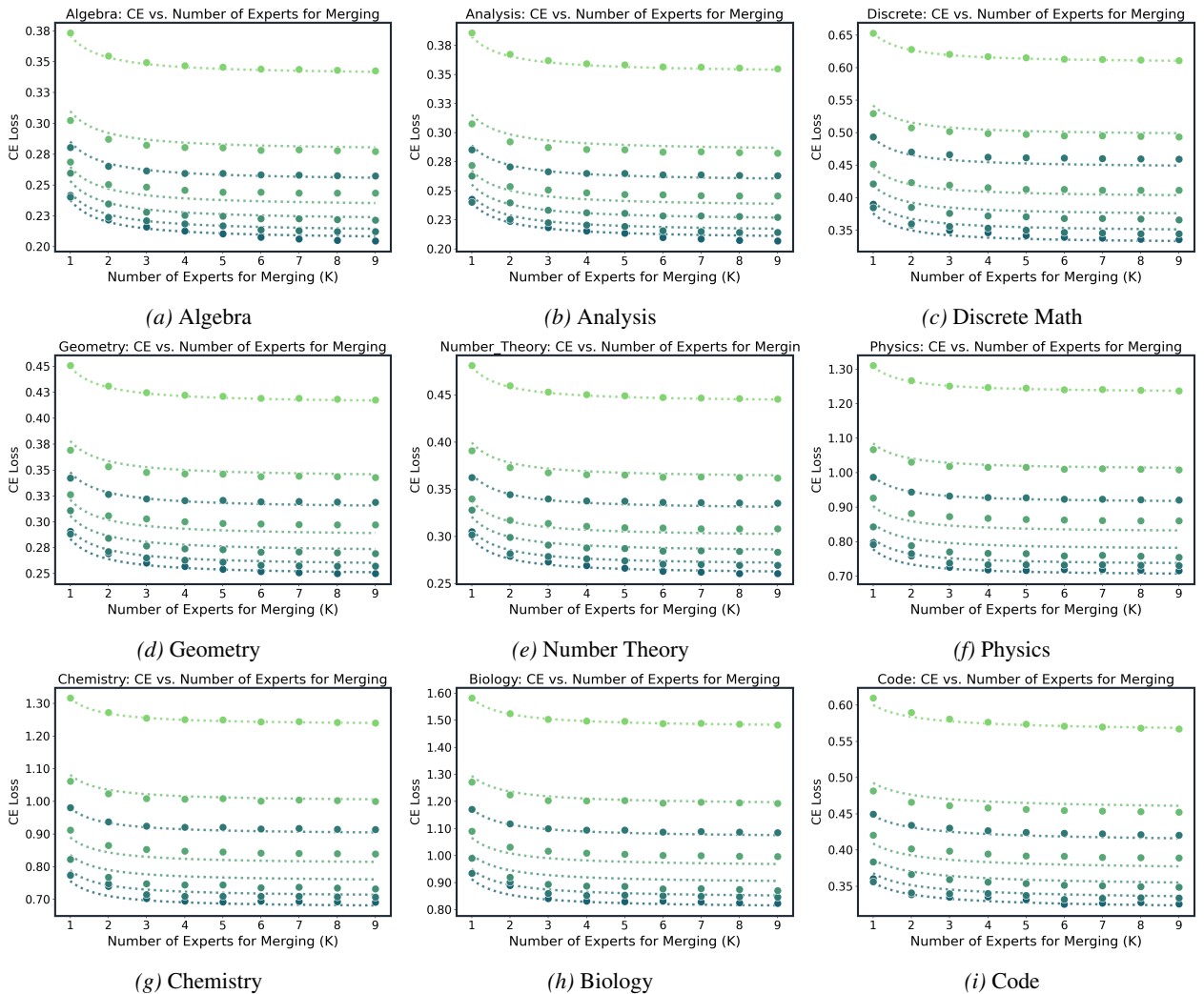

*Figure 17.* Merging Scaling Law with the TA Method

*Table 3.* Joint $(N, k)$ fit for Average (per-domain parameters).

| domain | Lstar | B | beta | A0 | gamma | b0 | R2 |
|--------|-------|---|------|-----|-------|-----|-----|
| algebra | 0.18092 | 0.11453 | 0.42335 | 0.052334 | 0.0086009 | 0.096378 | 0.984 |
| analysis | 0.18784 | 0.11445 | 0.46899 | 0.054877 | 0.02738 | 0.1375 | 0.988 |
| biology | 0.63884 | 0.6201 | 0.37247 | 0.1588 | 1.4702e-11 | 0.022561 | 0.990 |
| chemistry | 0.50824 | 0.54954 | 0.34262 | 0.12219 | 2.15e-08 | 1.668e-14 | 0.990 |
| code | 0.28292 | 0.20851 | 0.41186 | 0.082102 | 0.13678 | 0.43453 | 0.986 |
| discrete | 0.2052 | 0.3295 | 0.26766 | 0.066181 | 4.7525e-12 | 9.8614e-05 | 0.992 |
| geometry | 0.20278 | 0.16029 | 0.35431 | 0.052369 | 1.3982e-12 | 0.0087202 | 0.987 |
| number_theory | 0.21726 | 0.16818 | 0.38339 | 0.055823 | 6.8172e-09 | 0.0070628 | 0.992 |
| physics | 0.54195 | 0.52847 | 0.33756 | 0.1111 | 0.0038941 | 9.3222e-17 | 0.987 |

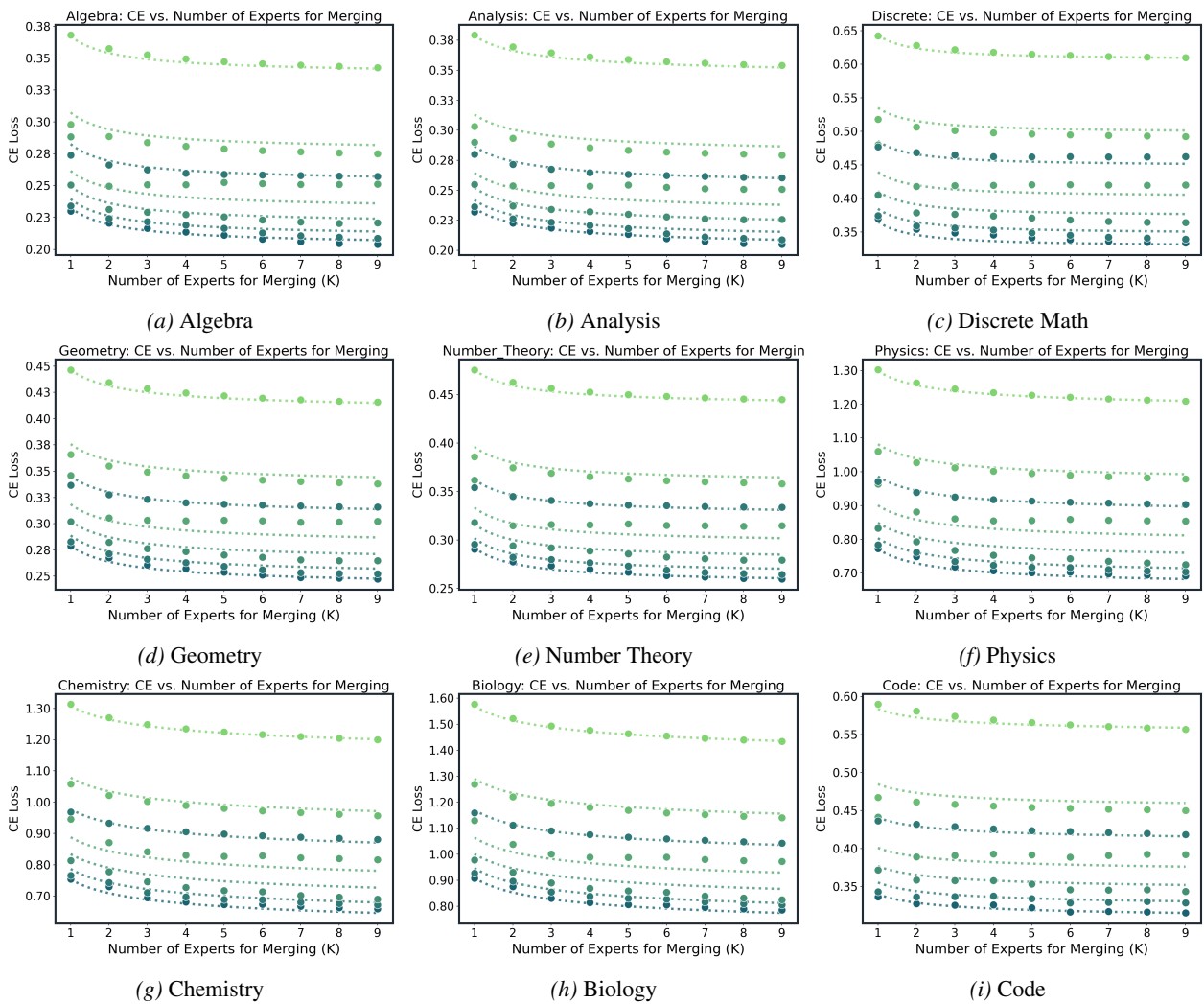

*Figure 18.* Merging Scaling Law with the TIES Method

*Table 4.* Joint $(N, k)$ fit for TA (per-domain parameters).

| domain | Lstar | B | beta | A0 | gamma | b0 | R2 |
|--------|-------|---|------|-----|-------|----|----|
| algebra | 0.1912 | 0.10481 | 0.48613 | 0.031756 | 2.0539e-12 | 3.0949e-12 | 0.993 |
| analysis | 0.19859 | 0.10452 | 0.53812 | 0.032072 | 0.020433 | 8.3408e-12 | 0.994 |
| biology | 0.67453 | 0.6048 | 0.39438 | 0.10437 | 2.0948e-10 | 6.7298e-13 | 0.994 |
| chemistry | 0.5471 | 0.52754 | 0.3698 | 0.079144 | 1.4331e-15 | 5.2296e-13 | 0.994 |
| code | 0.29195 | 0.19378 | 0.4604 | 0.061683 | 0.11845 | 0.41132 | 0.993 |
| discrete | 0.26439 | 0.26479 | 0.36064 | 0.045787 | 5.5863e-10 | 1.153e-15 | 0.997 |
| geometry | 0.21888 | 0.14605 | 0.40757 | 0.034849 | 3.6096e-12 | 6.4127e-08 | 0.995 |
| number_theory | 0.23532 | 0.15 | 0.45207 | 0.037155 | 2.7958e-12 | 4.9617e-11 | 0.997 |
| physics | 0.57646 | 0.50399 | 0.36559 | 0.073691 | 1.0052e-07 | 5.0247e-15 | 0.993 |

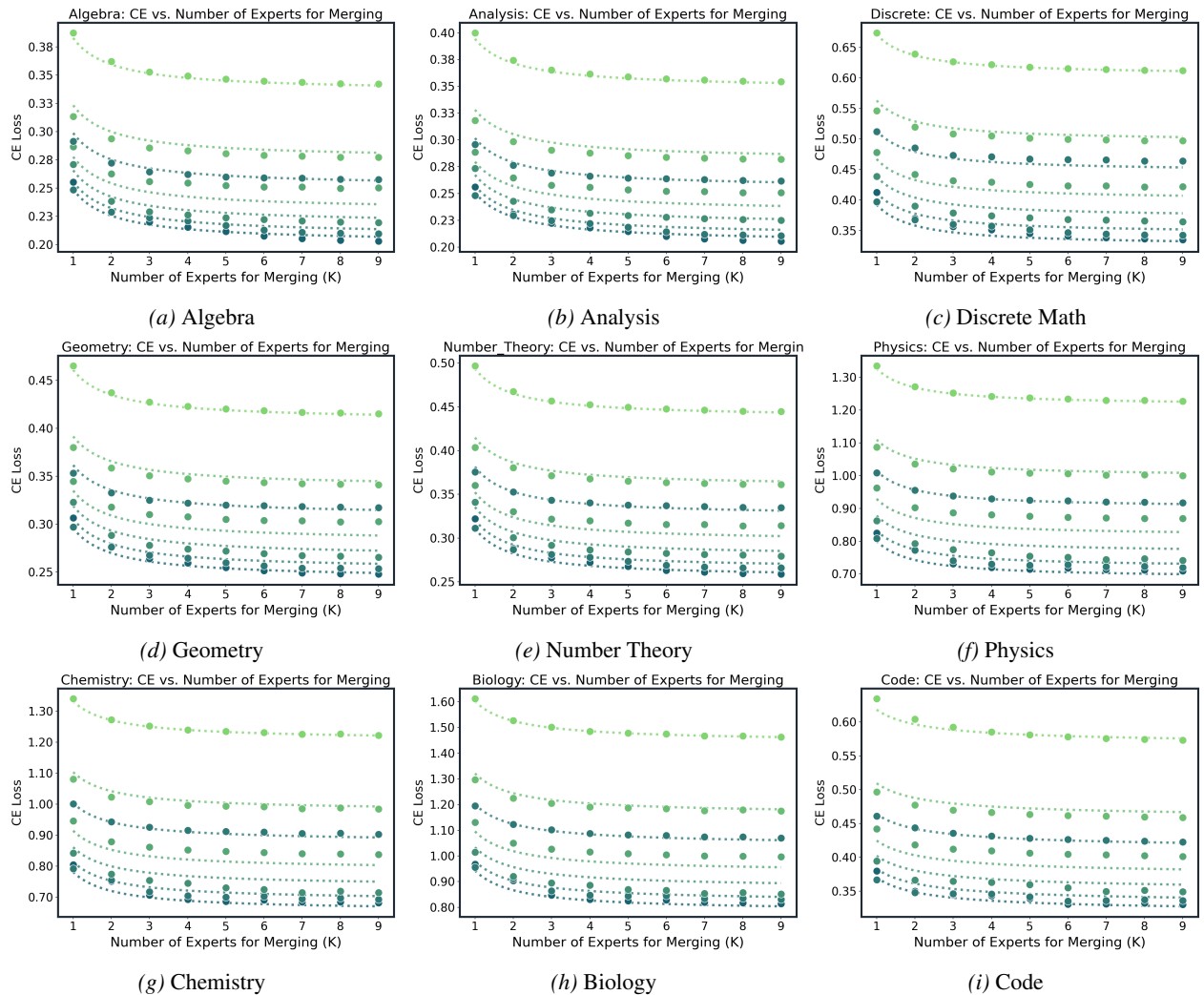

*Figure 19.* Merging Scaling Law with the DARE Method

*Table 5.* Joint $(N, k)$ fit for TIES (per-domain parameters).

| domain | Lstar | B | beta | A0 | gamma | b0 | R2 |
|---|---|---|---|---|---|---|---|
| algebra | 0.18929 | 0.10752 | 0.46554 | 0.035371 | 0.011425 | 0.19757 | 0.975 |
| analysis | 0.19237 | 0.10912 | 0.50434 | 0.050902 | 0.016536 | 0.58856 | 0.980 |
| biology | 0.6077 | 0.60414 | 0.38384 | 0.37301 | 0.0080666 | 1.1634 | 0.990 |
| chemistry | 0.48423 | 0.53452 | 0.35563 | 0.30644 | 0.0069314 | 1.2221 | 0.989 |
| code | 0.26877 | 0.21391 | 0.38297 | 0.079839 | 0.11961 | 1.1999 | 0.986 |
| discrete | 0.22555 | 0.30917 | 0.28993 | 0.037062 | 2.9507e-10 | 1.2161e-08 | 0.988 |
| geometry | 0.21017 | 0.15222 | 0.38672 | 0.051128 | 5.5086e-10 | 0.39637 | 0.983 |
| number_theory | 0.22585 | 0.15954 | 0.41453 | 0.046348 | 1.0173e-09 | 0.27291 | 0.987 |
| physics | 0.53415 | 0.51524 | 0.34897 | 0.15923 | 0.00073252 | 0.50358 | 0.987 |

## H.2. Variance: Joint $(N, k)$ Fits by Method

We fit $\mathrm{Var}[L_d \mid N, k] = V_{*,d} + B_d^{(\mathrm{var})} N^{-\beta_d^{(\mathrm{var})}} + \frac{A_{0,d}^{(\mathrm{var})} N^{-\gamma_d^{(\mathrm{var})}}}{k + b_{0,d}^{(\mathrm{var})}}$ with $V_{*,d} \approx 0$. Below we list parameters and $N{=}72\mathrm{B}$ predictions for $k \in \{1, 3, 5, 9\}$ in Tables 6–11.

*Table 6.* Variance fit parameters, Average.

| domain | ls | b | beta | a0 | gamma | b0 | r2 |
|---|---|---|---|---|---|---|---|
| algebra | 1.36e-18 | 5.57e-19 | 3 | 0.00159 | 0.0178 | 1.89e-11 | 0.844 |
| discrete | 5.80e-34 | 1.23e-22 | 1.94 | 0.00254 | 0.00496 | 2.29e-19 | 0.862 |
| analysis | 8.12e-29 | 2.49e-20 | 3 | 0.00146 | 0.0283 | 1.57e-12 | 0.861 |
| geometry | 2.30e-27 | 9.82e-22 | 2.1 | 0.00192 | 0.032 | 1.82e-16 | 0.851 |
| code | 3.25e-23 | 2.74e-22 | 0.067 | 0.0085 | 0.254 | 0.912 | 0.782 |
| number_theory | 2.92e-20 | 1.53e-23 | 2.08e-07 | 0.00248 | 0.0273 | 5.79e-13 | 0.86 |
| chemistry | 7.08e-31 | 9.84e-24 | 1.99 | 0.0393 | 0.127 | 0.205 | 0.891 |
| physics | 5.15e-21 | 1.09e-27 | 1.76 | 0.0229 | 0.119 | 0.125 | 0.903 |
| biology | 1.43e-19 | 1.77e-33 | 1.9 | 0.0556 | 0.151 | 0.272 | 0.879 |

*Table 7.* Variance at $N{=}72\mathrm{B}$, Average.

| domain | $k{=}1$ | $k{=}3$ | $k{=}5$ | $k{=}9$ |
|---|---|---|---|---|
| algebra | 0.00148 | 4.93e-04 | 2.96e-04 | 1.64e-04 |
| analysis | 0.0013 | 4.32e-04 | 2.59e-04 | 1.44e-04 |
| biology | 0.0229 | 0.00889 | 0.00552 | 0.00314 |
| chemistry | 0.0189 | 0.00711 | 0.00438 | 0.00247 |
| code | 0.0015 | 7.34e-04 | 4.86e-04 | 2.90e-04 |
| discrete | 0.00249 | 8.30e-04 | 4.98e-04 | 2.77e-04 |
| geometry | 0.00168 | 5.59e-04 | 3.35e-04 | 1.86e-04 |
| number_theory | 0.0022 | 7.34e-04 | 4.41e-04 | 2.45e-04 |
| physics | 0.0122 | 0.00441 | 0.00269 | 0.00151 |

*Table 8.* Variance fit parameters, TA.

| domain | ls | b | beta | a0 | gamma | b0 | r2 |
|---|---|---|---|---|---|---|---|
| algebra | 7.61e-31 | 9.32e-22 | 4.05e-08 | 0.00109 | 1.62e-06 | 3.41e-16 | 0.821 |
| discrete | 5.05e-36 | 9.52e-33 | 0.784 | 0.0017 | 1.14e-10 | 9.33e-23 | 0.819 |
| analysis | 2.63e-44 | 2.67e-26 | 0.585 | 9.98e-04 | 2.01e-06 | 7.55e-23 | 0.84 |
| geometry | 8.01e-30 | 4.73e-21 | 3.03e-08 | 0.00133 | 0.00682 | 5.56e-16 | 0.848 |
| code | 1.78e-33 | 2.30e-05 | 3 | 0.00525 | 0.206 | 0.514 | 0.816 |
| number_theory | 4.23e-45 | 1.72e-27 | 1.62 | 0.00171 | 5.25e-07 | 8.13e-21 | 0.845 |
| chemistry | 1.75e-24 | 3.78e-21 | 2.18 | 0.0266 | 0.112 | 1.88e-11 | 0.93 |
| physics | 6.05e-20 | 2.15e-21 | 0.812 | 0.0169 | 0.0995 | 1.70e-28 | 0.936 |
| biology | 2.27e-19 | 2.19e-23 | 1.91 | 0.0372 | 0.137 | 4.71e-12 | 0.924 |

*Table 9.* Variance at $N{=}72\mathrm{B}$, TA.

| domain | $k{=}1$ | $k{=}3$ | $k{=}5$ | $k{=}9$ |
|---|---|---|---|---|
| algebra | 0.00109 | 3.64e-04 | 2.19e-04 | 1.21e-04 |
| analysis | 9.98e-04 | 3.33e-04 | 2.00e-04 | 1.11e-04 |
| biology | 0.0207 | 0.00691 | 0.00414 | 0.0023 |
| chemistry | 0.0165 | 0.0055 | 0.0033 | 0.00183 |
| code | 0.00144 | 6.18e-04 | 3.94e-04 | 2.28e-04 |
| discrete | 0.0017 | 5.66e-04 | 3.40e-04 | 1.89e-04 |
| geometry | 0.0013 | 4.32e-04 | 2.59e-04 | 1.44e-04 |
| number_theory | 0.00171 | 5.70e-04 | 3.42e-04 | 1.90e-04 |
| physics | 0.011 | 0.00368 | 0.00221 | 0.00123 |

*Table 10.* Variance fit parameters, TIES.

| domain | ls | b | beta | a0 | gamma | b0 | r2 |
|---|---|---|---|---|---|---|---|
| algebra | 1.35e-27 | 4.08e-32 | 0.863 | 7.48e-04 | 7.09e-11 | 1.94e-09 | 0.801 |
| discrete | 2.48e-34 | 9.61e-29 | 2.98 | 0.00117 | 6.34e-13 | 8.13e-11 | 0.736 |
| analysis | 2.03e-22 | 1.96e-26 | 4.28e-08 | 6.83e-04 | 1.33e-08 | 3.31e-09 | 0.822 |
| geometry | 2.56e-27 | 6.97e-26 | 0.63 | 8.99e-04 | 1.43e-11 | 2.86e-08 | 0.784 |
| code | 2.92e-12 | 1.76e-05 | 3 | 0.00424 | 0.12 | 1.09 | 0.752 |
| number_theory | 2.25e-24 | 3.61e-33 | 1.11e-07 | 0.00114 | 8.10e-12 | 6.51e-11 | 0.796 |
| chemistry | 3.25e-49 | 2.78e-28 | 0.781 | 0.0241 | 0.00132 | 0.446 | 0.816 |
| physics | 2.45e-19 | 5.40e-27 | 3 | 0.0137 | 0.00363 | 0.164 | 0.886 |
| biology | 1.65e-23 | 9.54e-22 | 0.00561 | 0.0344 | 0.0366 | 0.397 | 0.857 |

*Table 11.* Variance at $N{=}72$B, TIES.

| domain | $k{=}1$ | $k{=}3$ | $k{=}5$ | $k{=}9$ |
|---|---|---|---|---|
| algebra | 7.48e-04 | 2.49e-04 | 1.50e-04 | 8.31e-05 |
| analysis | 6.83e-04 | 2.28e-04 | 1.37e-04 | 7.59e-05 |
| biology | 0.0211 | 0.00867 | 0.00546 | 0.00313 |
| chemistry | 0.0165 | 0.00694 | 0.00439 | 0.00253 |
| code | 0.00121 | 6.20e-04 | 4.17e-04 | 2.51e-04 |
| discrete | 0.00117 | 3.91e-04 | 2.35e-04 | 1.30e-04 |
| geometry | 8.99e-04 | 3.00e-04 | 1.80e-04 | 9.99e-05 |
| number_theory | 0.00114 | 3.81e-04 | 2.28e-04 | 1.27e-04 |
| physics | 0.0116 | 0.00426 | 0.00261 | 0.00147 |

# I. Cross-Domain Fit Details

### I.0.1. CROSS-DOMAIN (POOLED EVIDENCE)

Macro-averaged CE over the nine domains follows the same floor+tail law $L_\infty(N) + A(N)/(k{+}b)$ as in-domain: curves are monotone with steep early gains and a short inverse tail; TA and TIES(0.5) show slightly faster early drops and gaps narrow with $k$. A small bounded non-monotonicity appears for TIES(1) at 3B and is captured by adding a small bounded term in the fit. Scaling with model size mirrors the in-domain trend: at fixed large $k$ (e.g., $k{=}9$, Average), pooled CE improves substantially from small to large bases, reflecting a lower floor and a smaller tail amplitude. Merge-to-merge variance contracts approximately as $1/k$ with smaller amplitude at larger $N$, and TIES/TA exhibit slightly lower variance than Average at small $k$; details and extended forecasts (including 72B) appear in Tables 6–11. For fitting, we regress the mean law per base size and method (setting the interference term to zero for monotone series) and fit the variance model unweighted; the scale parameter decreases with $N$ and the variance floor is small. Representative figures include Average@32B (mean improving from 0.5173 to 0.4530; variance shrinking from $9.8{\times}10^{-4}$ to $4.3{\times}10^{-5}$), TIES(0.5)@14B (mean $0.5286{\to}0.4599$), and the bounded non-monotonicity for TIES(1)@3B captured by a positive interference term.

## I.1. Variance Behavior (Both Settings)

*Merge-to-merge variability* contracts approximately as

$$\text{Var}[L] \;\propto\; \frac{1}{k}, \tag{22}$$

with three robust regularities: (i) a near-$1/k$ drop that is already pronounced by small $k$ and flattens near a small floor (e.g., chemistry @0.5B, Average: $0.0385{\to}0.00108$ by $k{=}8$; algebra @0.5B: $2.28{\times}10^{-3}{\to}1.88{\times}10^{-5}$); (ii) *larger models are stabler*—at fixed $k$, variance is lower for larger $N$ (e.g., physics, Average, $k{=}1$: $0.0239{\to}0.0128$ from 0.5B to 32B); and (iii) *method ordering* at small $k$ typically satisfies TIES $<$ TA $<$ Average, with gaps vanishing as $k$ grows. We use Eq. equation 22 descriptively (fixing the log–log slope near $-1$), as heavier parameterization yields little additional predictive value while the simple form transfers cleanly across domains, sizes, and methods.

# J. Core Questions

## J.1. Per-domain fits, $k_\varepsilon$ examples, and robustness

**Specification.** For each domain $d$ we fit the joint law $\mathbb{E}[L_d \mid N, k] = L_{*,d} + B_d N^{-\beta_d} + \frac{A_{0,d} N^{-\gamma_d}}{k+b_d}$ by weighted nonlinear least squares (weights $\propto k$). We summarize *floors* via $L_{\infty,d}(N) = L_{*,d} + B_d N^{-\beta_d}$ (log–log regression) and *tails* via $A_d(N) = A_{0,d} N^{-\gamma_d}$.

**Per-domain parameters.** Floors exhibit tight power-law fits with exponents clustered in $[0.33, 0.42]$ and $R^2 \approx 0.98$–$0.99$ across domains. Tails are smaller and noisier; several domains are near-flat in $N$, while *code* shows the clearest decay. Table 12 lists an illustrative subset; full tables for all methods/domains appear in Appendix H.

| Domain | $\hat{b}$ | $\hat{A}_0$ | $\hat{\gamma}$ | $R^2(A)$ | $\hat{L}_*$ | $\hat{B}$ | $\hat{\beta}$ | $R^2(L)$ |
|---|---|---|---|---|---|---|---|---|
| algebra | 0.000 | 0.0460 | $-0.004$ | $-0.002$ | 0.1724 | 0.1248 | 0.379 | 0.983 |
| analysis | 0.000 | 0.0462 | $+0.009$ | $+0.009$ | 0.1793 | 0.1255 | 0.417 | 0.990 |
| biology | 0.125 | 0.1741 | $-0.006$ | $+0.007$ | 0.6227 | 0.6338 | 0.362 | 0.988 |
| chemistry | 0.075 | 0.1317 | $-0.006$ | $+0.008$ | 0.4924 | 0.5639 | 0.331 | 0.988 |
| code | 0.250 | 0.0682 | $+0.115$ | 0.556 | 0.2705 | 0.2238 | 0.378 | 0.986 |

*Table 12.* **Joint $(N, k)$ fits (subset, Average).** Floors are tight power laws; tails are small and domain-dependent (clearest decay in *code*).

**Macro evidence.** At $k=9$ (Average), macro CE decreases from 0.739 at 0.5B to 0.430 at 32B (–41.9%), consistent with a lower floor and a weakly shrinking tail.

$k_\varepsilon$ **examples ($\varepsilon=0.01$).** Using $k_\varepsilon(N, d) = \lceil A_d(N)/\varepsilon - b_d \rceil$ with $A_d(N) = A_{0,d} N^{-\gamma_d}$:

- *code*: $(\hat{b}, \hat{A}_0, \hat{\gamma}) = (0.25, 0.068, 0.115)$ gives $A(0.5B) \approx 0.074$, $A(32B) \approx 0.046$, hence $k_\varepsilon(0.5B) = 8$ and $k_\varepsilon(32B) = 5$.

- *biology*: $(0.125, 0.174, -0.006)$ (near-flat tail) gives $A(0.5B) \approx 0.173$, $A(32B) \approx 0.177$, so $k_\varepsilon$ stays $\approx 18$, yet CE still falls with $N$ due to the lower floor.

**Robustness.** Altering weights (uniform vs. $\propto k$) or censoring tiny high-$k$ points barely changes floor exponents. For extrapolation (e.g., 72B), floors should be treated as the dominant $N$-driver, with tails weakly decreasing/flat; $k_\varepsilon$ then gives a practical "experts-to-saturation" budget.

**Plot inventory.** (i) Macro CE@$k=9$ vs. $N$ (log–log) with power-law fit; (ii) two representative floor curves $L_{\infty,d}(N)$ (e.g., *algebra* vs. *biology*); (iii) optional $A_d(N)$ vs. $N$ to visualize tail trends.

## J.2. Most of the gain comes from the first few experts

We quantify the "return" from merging $k$ experts at a fixed $(N, d)$ by the fraction of realized improvement $R(N, d, k)$ computed from the monotone envelope of the measured CE curve (see Appendix J.2). We summarize two views in Fig. 20: *(left)* the median $R(k)$ over all $(N, d)$ with an IQR band; *(right)* a heatmap of the smallest $k$ that reaches a target return (here 90%). As shown in Fig. 20, most of the improvement arrives early: the median curve crosses 85% by $k=5$ and 90% by $k=6$, and the $k_{90}$ heatmap concentrates in $\{5, 6\}$ across domains and model sizes. Math-like domains tend to saturate slightly earlier, while science-like domains keep a longer—but still flattening—tail. This "early elbow" follows directly from the unified law $L(N, k) = L_\infty(N) + A(N)/(k+b)$: the marginal gain $\Delta_k \approx A(N)/[(k+b)(k+1+b)]$ decays roughly as $k^{-2}$, so returns diminish sharply beyond the first few experts.

## J.3. Additional plots, tables, and details

For each method (Average, TA, TIES, DARE) and size $N$, we fit $L(N, k) = L_\infty(N) + A(N)/(k+b)$ by weighted least squares (weights $\propto k$) on the pooled CE, averaging over randomized expert orders; only TIES with the strongest nonlinearity requires an extra bounded term $+D(N)\frac{k}{k+q}$, with small $D$ and stable $q$. We release per-method parameter

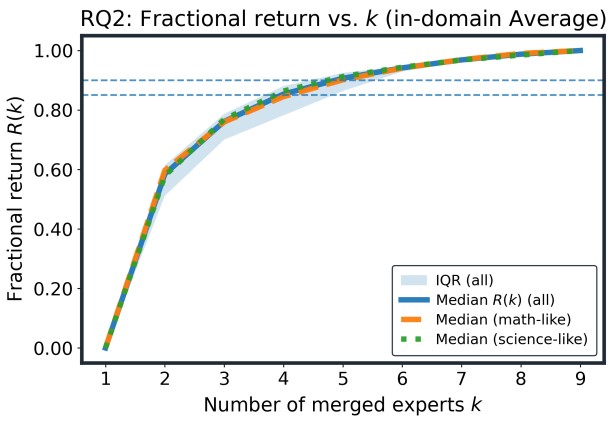
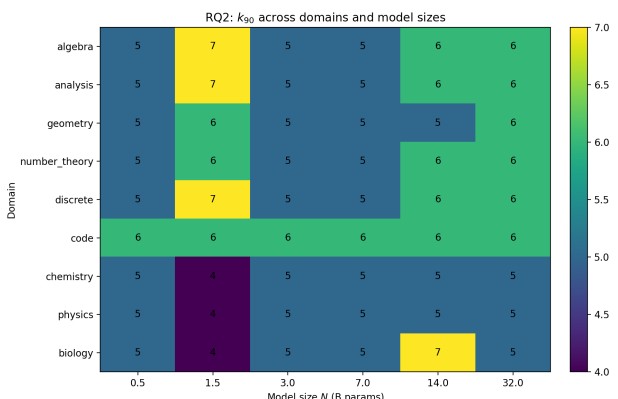

*Figure 20.* **Most of the gain comes from the first few experts.** *Left:* Median fractional return $R(k)$ with IQR band; $k=5$ and $k=6$ cross the 85%/90% thresholds, respectively. *Right:* $k_{90}$ across domains and sizes concentrates at $k \in \{5, 6\}$ (about half to two-thirds of this 9-expert pool ($5/9 \approx 56\%$)).

tables $\{b, A_0, \gamma, L_*, B, \beta\}$ and residual plots versus $k$; companion figures reproduce the floor/tail summaries in Fig. 4a for all methods and provide fractional-return curves $R(k)$ and $k_{90}$ heatmaps across $N$. Headline patterns match the main text: most pooled improvement is realized by $k \leq 6$, method differences narrow with $k$, and scaling in $N$ lowers both the pooled floor and the tail.

| Method | Qwen-0.5b | Qwen-1.5b | Qwen-3b | Qwen-7b | Qwen-14b | Qwen-32b | Qwen-72b | SUM(GPUh) |
|--------|-----------|-----------|---------|---------|----------|----------|----------|-----------|
| TA | 32s | 68s | 129s | 244s | 383s | 777s | 2686s | 1.20 |
| AVG | 48s | 73s | 168s | 265s | 421s | 843s | 2280s | 1.14 |
| Dare | 30s | 72s | 102s | 251s | 420s | 796s | 2360s | 1.112 |
| Ties | 43s | 77s | 136s | 270s | 507s | 961s | 2967s | 1.38 |

*Table 13.* GPU hours required to merge nine domains across model sizes.

| Model Size | Model Name |
|------------|------------|
| 3B | theprint/ReWiz-Llama-3.2-3B (theprint, 2025) |
| | NousResearch/Hermes-3-Llama-3.2-3B (NousResearch, 2025b) |
| | MergeBench/Llama-3.2-3B-Instruct_coding (MergeBench, 2025a) |
| | MergeBench/Llama-3.2-3B-Instruct_math (MergeBench, 2025c) |
| | MergeBench/Llama-3.2-3B-Instruct_multilingual (MergeBench, 2025d) |
| | meta-llama/Llama-3.2-3B-Instruct (meta llama, 2025b) |
| | ValiantLabs/Llama3.2-3B-ShiningValiant2 (ValiantLabs, 2025) |
| | MergeBench/Llama-3.2-3B-Instruct_safety (MergeBench, 2025e) |
| | MergeBench/Llama-3.2-3B-Instruct_instruction (MergeBench, 2025b) |
| 8B | Undi95/Meta-Llama-3-8B-Instruct-hf (Undi95, 2025b) |
| | Undi95/Llama-3-LewdPlay-8B-evo (Undi95, 2025a) |
| | jondurbin/bagel-8b-v1.0 (jondurbin, 2025) |
| | Weyaxi/Einstein-v6.1-Llama3-8B (Weyaxi, 2025) |
| | VAGOsolutions/Llama-3-SauerkrautLM-8b-Instruct (VAGOsolutions, 2025) |
| | aaditya/OpenBioLLM-Llama3-8B (aaditya, 2025) |
| | Dampfinchen/Llama-3-8B-Ultra-Instruct (Dampfinchen, 2025) |
| | NousResearch/Hermes-3-Llama-3.1-8B (NousResearch, 2025a) |
| | meta-llama/Llama-3.1-8B-Instruct (meta llama, 2025a) |

*Table 14.* List of open source models on Huggingface.

**Extended evidence**

*Small-k mean gaps vs. Average (relative %).* We report $(\mathrm{Avg} - \mathrm{Method})/\mathrm{Avg}$ at $k{=}2$ and the signed gap at $k{=}9$ (lower is better).

| | **0.5B** | | **14B** | |
|---|---|---|---|---|
| Method | $k{=}2$ | $k{=}9$ | $k{=}2$ | $k{=}9$ |
| TA ($\lambda{=}0.8$) | 0.9% | +0.7% (worse) | 0.6% | +1.4% (worse) |
| TIES ($\lambda{=}0.5$) | 0.9% | −1.1% (better) | 0.6% | −2.2% (better) |

| | **32B** | |
|---|---|---|
| Method | $k{=}2$ | $k{=}9$ |
| TA ($\lambda{=}0.8$) | 1.2% | +1.2% (worse) |
| TIES ($\lambda{=}0.5$) | 1.7% | −2.3% (better) |

*Summary.* The method "bandwidth" is consistently narrow across scales: at small $k$, TA and TIES(0.5) are modestly better than Average (typically $1\%\sim2\%$ at $k{=}2$), and by $k{=}9$ gaps further compress (TIES(0.5) usually retains a $\approx1\%\sim2\%$ edge, TA is near-tied or slightly worse). Variance shows the same convergence: at $N{=}32$B, $k{=}2$ the across-merge variance is $9.67{\times}10^{-4}$ (Average), $7.83{\times}10^{-4}$ (TA, $-19\%$), and $6.50{\times}10^{-4}$ (TIES 0.5, $-33\%$); by $k{=}8$ all methods are around $(3{-}4)\times10^{-5}$. At $N{=}0.5$B, $k{=}2$ the pattern holds (Average $1.73{\times}10^{-3}$, TA $-26\%$, TIES 0.5 $-44\%$). A mild bounded non-monotonicity for TIES($\lambda{=}1$) at 3B is captured by a small term $D\,\frac{k}{k+q}$; using $\lambda{=}0.5$ restores the standard monotone $1/(k+b)$ tail. Together, these results support the main-text claim: method differences are second-order and shrink quickly with $k$.

## K. Do Downstream Metrics Follow the Same Trend?

### K.1. Overall Results

**Setup.** To test whether the CE scaling law is reflected in end-task quality, we train and use the different merged checkpoints from Section 3 to demonstrate the trend on different backbones. In this section, we post-train on Llama-based and Gemma-based models and evaluate them on a diverse suite of downstream benchmarks, including math, general reasoning, multilingual, coding, and safety. For each backbone and merge method, we:

(i) evaluate all expert subsets for $k\in\{1,\ldots,5\}$,

(ii) normalise each metric so that larger is better,

(iii) report the *mean accuracy* obtained by first averaging across tasks and then across all expert subsets at fixed $k$.

Table 15 summarises the resulting trend for three backbones (LLaMA-3.1 8B, LLaMA-3.2 3B, Gemma-2 2B) and two merge rules (Task Arithmetic and TIES).

**Findings.** Across all settings, aggregated downstream performance generally improves as we increase the number of merged experts and then saturates, consistent with the same diminishing-return intuition observed for CE, although the plateau often appears earlier. ... For LLaMA-3.1 8B with Task Arithmetic, mean accuracy rises steadily from 0.41 at $k{=}1$ to 0.47 at $k{=}5$, with rapidly diminishing gains after $k{\approx}3$. LLaMA-3.2 3B shows the same qualitative pattern but with a shallower tail: accuracy improves from 0.38 at $k{=}2$ to about 0.39 at $k{=}4$ and then slightly fluctuates within $\pm0.002$, which we attribute to benchmark variance rather than a systematic degradation. Gemma-2 2B (available only for $k{\geq}2$) and TIES on LLaMA-3.1 8B both display monotone or nearly monotone gains up to $k{\approx}4$, followed by a clear plateau. Taken together, these results suggest that the CE scaling law is informative about the practical utility regime of merging, while downstream metrics remain complementary and noisier indicators rather than quantities we claim to follow the same parametric law. A more refined characterisation of when CE and task accuracy may diverge, and how to predict in advance whether a particular merge will work on a given task, is an interesting direction for future work.

| Backbone | Method | $k=1$ | $k=2$ | $k=3$ | $k=4$ | $k=5$ |
|---|---|---|---|---|---|---|
| LLaMA-3.1 8B | TA | 0.411 | 0.443 | 0.456 | 0.462 | 0.469 |
| LLaMA-3.2 3B | TA | 0.375 | 0.386 | 0.388 | 0.389 | 0.388 |
| Gemma-2 2B | TA | 0.492 | 0.503 | 0.506 | 0.507 | 0.507 |
| LLaMA-3.1 8B | TIES | 0.388 | 0.414 | 0.426 | 0.436 | 0.436 |

*Table 15.* Mean downstream accuracy vs. number of merged experts $k$. Each entry is averaged over all benchmarks and all expert subsets at fixed $k$ for the given backbone and merge methods (higher is better).

| folder | math_500 | gsm8k | ifeval | arc | hellaswag | mmlu | mbppplus | humanevalplus | wildguard_micro_harm | harmbench_micro_asr | wildguard_rta | harmbench_rta |
|---|---|---|---|---|---|---|---|---|---|---|---|---|
| 1 | 0.138 | 0.378 | 0.244 | 0.406 | 0.493 | 0.500 | 0.558 | 0.543 | 0.471 | 0.669 | 0.529 | 0.331 |
| 2 | 0.468 | 0.835 | 0.102 | 0.341 | 0.466 | 0.451 | 0.370 | 0.256 | 0.565 | 0.634 | 0.435 | 0.366 |
| 3 | 0.150 | 0.271 | 0.545 | 0.412 | 0.494 | 0.501 | 0.526 | 0.372 | 0.506 | 0.550 | 0.494 | 0.450 |
| 4 | 0.016 | 0.148 | 0.198 | 0.357 | 0.467 | 0.434 | 0.413 | 0.226 | 0.228 | 0.184 | 0.772 | 0.816 |
| 5 | 0.128 | 0.215 | 0.165 | 0.432 | 0.496 | 0.540 | 0.540 | 0.085 | 0.485 | 0.584 | 0.515 | 0.416 |
| 1-2 | 0.406 | 0.804 | 0.189 | 0.380 | 0.492 | 0.499 | 0.526 | 0.494 | 0.558 | 0.650 | 0.442 | 0.350 |
| 1-3 | 0.168 | 0.539 | 0.368 | 0.440 | 0.510 | 0.527 | 0.563 | 0.549 | 0.523 | 0.719 | 0.477 | 0.281 |
| 1-4 | 0.116 | 0.444 | 0.189 | 0.393 | 0.493 | 0.499 | 0.534 | 0.506 | 0.191 | 0.138 | 0.809 | 0.863 |
| 1-5 | 0.148 | 0.493 | 0.231 | 0.442 | 0.509 | 0.537 | 0.550 | 0.396 | 0.547 | 0.675 | 0.453 | 0.325 |
| 2-3 | 0.442 | 0.819 | 0.268 | 0.378 | 0.495 | 0.497 | 0.511 | 0.354 | 0.557 | 0.653 | 0.443 | 0.347 |
| 2-4 | 0.348 | 0.732 | 0.190 | 0.365 | 0.483 | 0.475 | 0.463 | 0.323 | 0.350 | 0.459 | 0.650 | 0.541 |
| 2-5 | 0.402 | 0.806 | 0.131 | 0.378 | 0.490 | 0.513 | 0.511 | 0.329 | 0.537 | 0.609 | 0.463 | 0.391 |
| 3-4 | 0.010 | 0.219 | 0.290 | 0.395 | 0.490 | 0.480 | 0.537 | 0.317 | 0.200 | 0.163 | 0.800 | 0.838 |
| 3-5 | 0.146 | 0.543 | 0.392 | 0.442 | 0.504 | 0.542 | 0.534 | 0.311 | 0.509 | 0.606 | 0.491 | 0.394 |
| 4-5 | 0.042 | 0.118 | 0.146 | 0.402 | 0.493 | 0.500 | 0.503 | 0.165 | 0.218 | 0.244 | 0.782 | 0.756 |
| 1-2-3 | 0.388 | 0.767 | 0.244 | 0.397 | 0.503 | 0.517 | 0.561 | 0.451 | 0.558 | 0.653 | 0.442 | 0.347 |
| 1-2-4 | 0.316 | 0.704 | 0.202 | 0.379 | 0.495 | 0.503 | 0.537 | 0.451 | 0.343 | 0.463 | 0.657 | 0.538 |
| 1-2-5 | 0.368 | 0.753 | 0.177 | 0.401 | 0.501 | 0.522 | 0.563 | 0.451 | 0.555 | 0.672 | 0.445 | 0.328 |
| 1-3-4 | 0.116 | 0.510 | 0.263 | 0.407 | 0.504 | 0.515 | 0.537 | 0.494 | 0.216 | 0.166 | 0.784 | 0.834 |
| 1-3-5 | 0.176 | 0.503 | 0.333 | 0.447 | 0.512 | 0.542 | 0.563 | 0.433 | 0.565 | 0.725 | 0.435 | 0.275 |
| 1-4-5 | 0.114 | 0.436 | 0.211 | 0.407 | 0.504 | 0.524 | 0.556 | 0.348 | 0.248 | 0.203 | 0.752 | 0.797 |
| 2-3-4 | 0.318 | 0.713 | 0.255 | 0.385 | 0.494 | 0.494 | 0.505 | 0.390 | 0.356 | 0.463 | 0.644 | 0.538 |
| 2-3-5 | 0.388 | 0.759 | 0.237 | 0.400 | 0.500 | 0.526 | 0.529 | 0.372 | 0.535 | 0.647 | 0.465 | 0.353 |
| 2-4-5 | 0.310 | 0.688 | 0.174 | 0.391 | 0.494 | 0.509 | 0.513 | 0.335 | 0.405 | 0.497 | 0.595 | 0.503 |
| 3-4-5 | 0.050 | 0.354 | 0.261 | 0.411 | 0.500 | 0.512 | 0.540 | 0.262 | 0.247 | 0.316 | 0.753 | 0.684 |
| 1-2-3-4 | 0.300 | 0.692 | 0.240 | 0.395 | 0.502 | 0.515 | 0.532 | 0.463 | 0.360 | 0.528 | 0.640 | 0.472 |
| 1-2-3-5 | 0.324 | 0.722 | 0.229 | 0.413 | 0.507 | 0.532 | 0.566 | 0.451 | 0.567 | 0.663 | 0.433 | 0.338 |
| 1-2-4-5 | 0.300 | 0.681 | 0.203 | 0.396 | 0.502 | 0.521 | 0.558 | 0.402 | 0.371 | 0.538 | 0.629 | 0.463 |
| 1-3-4-5 | 0.122 | 0.468 | 0.277 | 0.416 | 0.508 | 0.532 | 0.556 | 0.360 | 0.268 | 0.256 | 0.732 | 0.744 |
| 2-3-4-5 | 0.292 | 0.674 | 0.240 | 0.400 | 0.499 | 0.516 | 0.534 | 0.378 | 0.399 | 0.513 | 0.601 | 0.488 |
| 1-2-3-4-5 | 0.272 | 0.679 | 0.251 | 0.408 | 0.505 | 0.528 | 0.569 | 0.421 | 0.386 | 0.572 | 0.614 | 0.428 |
| overall_k1 | 0.180 | 0.369 | 0.251 | 0.390 | 0.483 | 0.485 | 0.481 | 0.296 | 0.451 | 0.524 | 0.549 | 0.476 |
| overall_k2 | 0.223 | 0.552 | 0.239 | 0.402 | 0.496 | 0.507 | 0.523 | 0.374 | 0.419 | 0.492 | 0.581 | 0.508 |
| overall_k3 | 0.254 | 0.619 | 0.235 | 0.403 | 0.501 | 0.516 | 0.540 | 0.399 | 0.403 | 0.480 | 0.597 | 0.520 |
| overall_k4 | 0.268 | 0.647 | 0.238 | 0.404 | 0.504 | 0.523 | 0.549 | 0.411 | 0.393 | 0.499 | 0.607 | 0.501 |
| overall_k5 | 0.272 | 0.679 | 0.251 | 0.408 | 0.505 | 0.528 | 0.569 | 0.421 | 0.386 | 0.572 | 0.614 | 0.428 |

*Table 16.* Full downstream results for LLaMA 3.1 8B with TA merging across five domain experts.

## K.2. Detailed Cases

**Setup.** We report full downstream results for three backbones and two merge rules. For all experiments, we start from a common base model and five domain-specialised experts: (1) *math* (MATH/GSM8K style), (2) *code* (MBPP/HumanEval style), (3) *multilingual* (general language understanding across languages), (4) *safety* (safety and refusal tuning), and (5) *instruction-following* (generic chat/IFEval). In the tables, the column `folder` encodes which experts are merged: for example, 1, 2, ..., 5 are single experts; `1-2` or `3-5` are 2-model merges of the corresponding experts; and `1-2-3-4-5` is the merge of all five experts. Tables 16 and 17 use *Task Arithmetic (TA)* as the merge rule on LLaMA 3.1 8B and LLaMA 3.2 3B, respectively. Table 18 uses TA on Gemma 2 2B. Table 19 uses *TIES* merging on LLaMA 3.1 8B. We evaluate on a heterogeneous benchmark suite covering math reasoning (math_500, GSM8K), code (mbppplus, humanevalplus), general QA and language understanding (IFEval, ARC, HellaSwag, MMLU and multilingual_overall), and safety. Rows `overall_k1–overall_k5` in the TA tables aggregate over all expert subsets with fixed $k$ and then average across benchmarks. These aggregated means are the values used in the main-text plots.

## L. Scaling Behaviour with 16 Domains

**Setup.** We extend the cross-domain scaling experiment to a larger 16-domain pool on the LLaMA3-3B-Instruct backbone. Starting from the original 9 domains (*algebra, analysis, geometry, discrete, number_theory, code, chemistry, physics, biology*), we add 7 additional experts fine-tuned on *Japanese, medical, house-arrangement, Korean, emotion, elementary*

| folder | math_500 | gsm8k | ifeval | arc | hellaswag | mmlu | multilingual_overall | mbppplus | humanevalplus | wildguard_micro_harm | harmbench_micro_asr | wildguard_rta | harmbench_rta |
|---|---|---|---|---|---|---|---|---|---|---|---|---|---|
| 1 | 0.048 | 0.256 | 0.177 | 0.349 | 0.437 | 0.458 | 0.415 | 0.466 | 0.415 | 0.607 | 0.756 | 0.393 | 0.244 |
| 2 | 0.274 | 0.682 | 0.196 | 0.329 | 0.423 | 0.444 | 0.398 | 0.386 | 0.287 | 0.625 | 0.656 | 0.375 | 0.344 |
| 3 | 0.070 | 0.194 | 0.351 | 0.350 | 0.431 | 0.425 | 0.402 | 0.410 | 0.293 | 0.567 | 0.663 | 0.433 | 0.338 |
| 4 | 0.002 | 0.036 | 0.161 | 0.330 | 0.429 | 0.454 | 0.404 | 0.423 | 0.220 | 0.146 | 0.109 | 0.854 | 0.891 |
| 5 | 0.016 | 0.061 | 0.187 | 0.356 | 0.429 | 0.441 | 0.409 | 0.421 | 0.213 | 0.697 | 0.784 | 0.303 | 0.216 |
| 1-2 | 0.192 | 0.545 | 0.189 | 0.347 | 0.432 | 0.457 | 0.412 | 0.442 | 0.335 | 0.637 | 0.713 | 0.363 | 0.288 |
| 1-3 | 0.096 | 0.261 | 0.340 | 0.361 | 0.442 | 0.455 | 0.420 | 0.429 | 0.360 | 0.595 | 0.691 | 0.405 | 0.309 |
| 1-4 | 0.076 | 0.240 | 0.214 | 0.344 | 0.435 | 0.460 | 0.413 | 0.455 | 0.305 | 0.427 | 0.513 | 0.573 | 0.488 |
| 1-5 | 0.058 | 0.217 | 0.194 | 0.359 | 0.437 | 0.459 | 0.418 | 0.471 | 0.372 | 0.690 | 0.713 | 0.310 | 0.288 |
| 2-3 | 0.184 | 0.519 | 0.263 | 0.350 | 0.436 | 0.451 | 0.413 | 0.397 | 0.305 | 0.619 | 0.681 | 0.381 | 0.319 |
| 2-4 | 0.124 | 0.519 | 0.185 | 0.336 | 0.430 | 0.455 | 0.407 | 0.434 | 0.262 | 0.526 | 0.631 | 0.474 | 0.369 |
| 2-5 | 0.098 | 0.434 | 0.185 | 0.351 | 0.430 | 0.456 | 0.412 | 0.426 | 0.280 | 0.705 | 0.709 | 0.295 | 0.291 |
| 3-4 | 0.066 | 0.110 | 0.316 | 0.347 | 0.439 | 0.451 | 0.412 | 0.402 | 0.274 | 0.294 | 0.375 | 0.706 | 0.625 |
| 3-5 | 0.046 | 0.064 | 0.266 | 0.361 | 0.439 | 0.447 | 0.415 | 0.426 | 0.280 | 0.654 | 0.663 | 0.346 | 0.338 |
| 4-5 | 0.020 | 0.104 | 0.218 | 0.349 | 0.432 | 0.455 | 0.412 | 0.431 | 0.195 | 0.509 | 0.513 | 0.491 | 0.488 |
| 1-2-3 | 0.160 | 0.467 | 0.257 | 0.357 | 0.439 | 0.457 | 0.418 | 0.447 | 0.372 | 0.623 | 0.731 | 0.377 | 0.269 |
| 1-2-4 | 0.126 | 0.447 | 0.194 | 0.344 | 0.434 | 0.461 | 0.413 | 0.429 | 0.305 | 0.565 | 0.675 | 0.435 | 0.325 |
| 1-2-5 | 0.096 | 0.417 | 0.179 | 0.353 | 0.434 | 0.459 | 0.415 | 0.423 | 0.335 | 0.668 | 0.700 | 0.332 | 0.300 |
| 1-3-4 | 0.068 | 0.246 | 0.279 | 0.354 | 0.440 | 0.457 | 0.417 | 0.434 | 0.317 | 0.442 | 0.528 | 0.558 | 0.472 |
| 1-3-5 | 0.052 | 0.218 | 0.259 | 0.360 | 0.442 | 0.457 | 0.419 | 0.434 | 0.323 | 0.670 | 0.653 | 0.330 | 0.347 |
| 1-4-5 | 0.050 | 0.207 | 0.216 | 0.351 | 0.436 | 0.459 | 0.416 | 0.442 | 0.262 | 0.590 | 0.650 | 0.410 | 0.350 |
| 2-3-4 | 0.134 | 0.399 | 0.251 | 0.347 | 0.436 | 0.455 | 0.413 | 0.418 | 0.280 | 0.489 | 0.519 | 0.511 | 0.481 |
| 2-3-5 | 0.094 | 0.340 | 0.248 | 0.359 | 0.438 | 0.455 | 0.417 | 0.402 | 0.329 | 0.684 | 0.669 | 0.316 | 0.331 |
| 2-4-5 | 0.078 | 0.352 | 0.203 | 0.345 | 0.432 | 0.459 | 0.412 | 0.423 | 0.287 | 0.630 | 0.706 | 0.370 | 0.294 |
| 3-4-5 | 0.038 | 0.080 | 0.285 | 0.353 | 0.438 | 0.454 | 0.415 | 0.418 | 0.244 | 0.475 | 0.525 | 0.525 | 0.475 |
| 1-2-3-4 | 0.106 | 0.409 | 0.242 | 0.350 | 0.438 | 0.459 | 0.416 | 0.442 | 0.323 | 0.542 | 0.594 | 0.458 | 0.406 |
| 1-2-3-5 | 0.090 | 0.394 | 0.227 | 0.360 | 0.439 | 0.459 | 0.419 | 0.439 | 0.335 | 0.666 | 0.666 | 0.334 | 0.334 |
| 1-2-4-5 | 0.086 | 0.372 | 0.214 | 0.349 | 0.435 | 0.461 | 0.415 | 0.452 | 0.317 | 0.636 | 0.688 | 0.364 | 0.313 |
| 1-3-4-5 | 0.040 | 0.204 | 0.250 | 0.355 | 0.440 | 0.458 | 0.418 | 0.431 | 0.329 | 0.563 | 0.597 | 0.437 | 0.403 |
| 2-3-4-5 | 0.084 | 0.272 | 0.226 | 0.352 | 0.437 | 0.457 | 0.416 | 0.405 | 0.280 | 0.598 | 0.597 | 0.402 | 0.403 |
| 1-2-3-4-5 | 0.070 | 0.334 | 0.231 | 0.356 | 0.438 | 0.459 | 0.418 | 0.423 | 0.305 | 0.602 | 0.644 | 0.398 | 0.356 |
| overall_k1 | 0.082 | 0.246 | 0.214 | 0.343 | 0.430 | 0.444 | 0.406 | 0.421 | 0.285 | 0.528 | 0.594 | 0.472 | 0.406 |
| overall_k2 | 0.096 | 0.301 | 0.237 | 0.350 | 0.435 | 0.455 | 0.413 | 0.431 | 0.297 | 0.566 | 0.620 | 0.434 | 0.380 |
| overall_k3 | 0.090 | 0.317 | 0.237 | 0.352 | 0.437 | 0.457 | 0.416 | 0.427 | 0.305 | 0.584 | 0.636 | 0.416 | 0.364 |
| overall_k4 | 0.081 | 0.330 | 0.232 | 0.353 | 0.438 | 0.459 | 0.417 | 0.434 | 0.317 | 0.601 | 0.628 | 0.399 | 0.372 |
| overall_k5 | 0.070 | 0.334 | 0.231 | 0.356 | 0.438 | 0.459 | 0.418 | 0.423 | 0.305 | 0.602 | 0.644 | 0.398 | 0.356 |

*Table 17.* Full downstream results for LLaMA 3.2 3B with TA merging across five domain experts.

*school mathematics*, and *Java code* tasks. For each domain, we merge $k \in \{2, 4, 6, 8, 10, 12, 14, 16\}$ experts using TA, sampling multiple random $k$-subsets of experts, and evaluating CE on the corresponding domain. We report the mean CE, together with the empirical variance and standard deviation across random subsets. The overall row represents the macro-average across all 16 domains for each $k$.

**Findings.** As shown in Table 20, CE decreases as the number of merged experts $k$ grows, both per-domain and in the 16-domain macro-average, with clear diminishing returns: most of the improvement is obtained by small $k$, and the gains flatten as $k$ increases from 10 to 16. At the same time, the empirical variance and standard deviation across random expert subsets shrink with $k$, indicating that the merging outcomes become more stable as more experts are combined. Crucially, moving from 9 to 16 domains does not change the qualitative behaviour. The aggregated CE in the 16-domain setting still exhibits the same floor+tail scaling in $k$ as in our main experiments.

| folder | math_500 | gsm8k | ifeval | arc | hellaswag | mmlu | mbppplus | humanevalplus | wildguard_rta | harmbench_rta |
|---|---|---|---|---|---|---|---|---|---|---|
| 1-2 | 0.288 | 0.569 | 0.417 | 0.372 | 0.431 | 0.488 | 0.437 | 0.366 | 0.826 | 0.806 |
| 1-3 | 0.254 | 0.566 | 0.457 | 0.378 | 0.432 | 0.488 | 0.447 | 0.348 | 0.824 | 0.859 |
| 1-4 | 0.240 | 0.560 | 0.440 | 0.378 | 0.431 | 0.488 | 0.437 | 0.335 | 0.837 | 0.884 |
| 1-5 | 0.264 | 0.531 | 0.463 | 0.386 | 0.437 | 0.490 | 0.447 | 0.354 | 0.817 | 0.828 |
| 2-4 | 0.290 | 0.591 | 0.451 | 0.373 | 0.432 | 0.487 | 0.442 | 0.335 | 0.797 | 0.825 |
| 3-4 | 0.302 | 0.590 | 0.421 | 0.373 | 0.431 | 0.487 | 0.437 | 0.354 | 0.814 | 0.841 |
| 2-3 | 0.276 | 0.569 | 0.442 | 0.381 | 0.434 | 0.488 | 0.439 | 0.348 | 0.812 | 0.834 |
| 2-5 | 0.258 | 0.591 | 0.449 | 0.376 | 0.432 | 0.487 | 0.434 | 0.305 | 0.828 | 0.856 |
| 3-5 | 0.266 | 0.557 | 0.468 | 0.386 | 0.437 | 0.490 | 0.431 | 0.329 | 0.817 | 0.856 |
| 4-5 | 0.252 | 0.550 | 0.451 | 0.386 | 0.437 | 0.489 | 0.450 | 0.341 | 0.840 | 0.866 |
| 1-2-4 | 0.292 | 0.558 | 0.438 | 0.374 | 0.433 | 0.489 | 0.442 | 0.366 | 0.817 | 0.831 |
| 1-2-3 | 0.298 | 0.544 | 0.407 | 0.371 | 0.432 | 0.488 | 0.452 | 0.372 | 0.836 | 0.847 |
| 1-2-5 | 0.286 | 0.536 | 0.438 | 0.384 | 0.435 | 0.489 | 0.434 | 0.378 | 0.820 | 0.819 |
| 1-3-4 | 0.268 | 0.553 | 0.453 | 0.377 | 0.433 | 0.487 | 0.455 | 0.360 | 0.842 | 0.897 |
| 1-3-5 | 0.266 | 0.538 | 0.444 | 0.388 | 0.439 | 0.490 | 0.434 | 0.348 | 0.809 | 0.863 |
| 1-4-5 | 0.264 | 0.522 | 0.449 | 0.385 | 0.438 | 0.489 | 0.434 | 0.360 | 0.833 | 0.888 |
| 2-3-4 | 0.298 | 0.575 | 0.451 | 0.374 | 0.433 | 0.487 | 0.447 | 0.348 | 0.813 | 0.863 |
| 2-3-5 | 0.298 | 0.557 | 0.444 | 0.382 | 0.437 | 0.489 | 0.450 | 0.354 | 0.802 | 0.822 |
| 2-4-5 | 0.260 | 0.560 | 0.449 | 0.381 | 0.437 | 0.489 | 0.444 | 0.348 | 0.818 | 0.847 |
| 3-4-5 | 0.256 | 0.538 | 0.470 | 0.383 | 0.438 | 0.490 | 0.452 | 0.354 | 0.828 | 0.888 |
| 1-2-3-4 | 0.282 | 0.557 | 0.423 | 0.375 | 0.433 | 0.488 | 0.437 | 0.360 | 0.833 | 0.875 |
| 1-2-3-5 | 0.286 | 0.543 | 0.458 | 0.384 | 0.438 | 0.489 | 0.452 | 0.335 | 0.809 | 0.834 |
| 1-2-4-5 | 0.282 | 0.547 | 0.438 | 0.380 | 0.437 | 0.490 | 0.444 | 0.354 | 0.828 | 0.856 |
| 1-3-4-5 | 0.280 | 0.528 | 0.442 | 0.388 | 0.439 | 0.489 | 0.444 | 0.341 | 0.838 | 0.903 |
| 2-3-4-5 | 0.288 | 0.563 | 0.470 | 0.380 | 0.439 | 0.489 | 0.447 | 0.354 | 0.833 | 0.863 |
| 1-2-3-4-5 | 0.270 | 0.542 | 0.457 | 0.381 | 0.439 | 0.489 | 0.439 | 0.341 | 0.826 | 0.872 |

*Table 18.* Full downstream results for Gemma 2 2B with TA merging across five domain experts.

| folder | math_500 | gsm8k | ifeval | arc | hellaswag | mmlu | mbppplus | humanevalplus | wildguard_rta | harmbench_rta | | |
|---|---|---|---|---|---|---|---|---|---|---|---|---|
| 1-2 | 0.264 | 0.604 | 0.190 | 0.408 | 0.486 | 0.542 | 0.545 | 0.402 | 0.579 | 0.421 | 0.688 | 0.312 |
| 1-3 | 0.090 | 0.418 | 0.205 | 0.421 | 0.487 | 0.544 | 0.526 | 0.335 | 0.610 | 0.390 | 0.747 | 0.253 |
| 1-4 | 0.110 | 0.466 | 0.233 | 0.406 | 0.488 | 0.541 | 0.529 | 0.335 | 0.387 | 0.613 | 0.463 | 0.537 |
| 1-5 | 0.080 | 0.418 | 0.233 | 0.416 | 0.485 | 0.542 | 0.529 | 0.311 | 0.614 | 0.386 | 0.734 | 0.266 |
| 2-4 | 0.084 | 0.418 | 0.205 | 0.392 | 0.483 | 0.531 | 0.511 | 0.299 | 0.431 | 0.569 | 0.497 | 0.503 |
| 3-4 | 0.084 | 0.418 | 0.205 | 0.398 | 0.482 | 0.533 | 0.521 | 0.244 | 0.393 | 0.607 | 0.422 | 0.578 |
| 2-3 | 0.210 | 0.604 | 0.203 | 0.398 | 0.481 | 0.538 | 0.545 | 0.305 | 0.590 | 0.410 | 0.716 | 0.284 |
| 2-5 | 0.064 | 0.137 | 0.246 | 0.389 | 0.479 | 0.535 | 0.497 | 0.287 | 0.595 | 0.405 | 0.719 | 0.281 |
| 3-5 | 0.064 | 0.137 | 0.246 | 0.406 | 0.481 | 0.536 | 0.521 | 0.244 | 0.558 | 0.442 | 0.766 | 0.234 |
| 4-5 | 0.066 | 0.359 | 0.196 | 0.393 | 0.480 | 0.531 | 0.508 | 0.250 | 0.403 | 0.597 | 0.466 | 0.534 |
| 1-2-4 | 0.246 | 0.596 | 0.216 | 0.396 | 0.488 | 0.539 | 0.540 | 0.341 | 0.458 | 0.542 | 0.500 | 0.500 |
| 1-2-3 | 0.248 | 0.609 | 0.194 | 0.411 | 0.487 | 0.544 | 0.524 | 0.372 | 0.582 | 0.418 | 0.697 | 0.303 |
| 1-2-5 | 0.242 | 0.600 | 0.194 | 0.403 | 0.485 | 0.541 | 0.537 | 0.305 | 0.575 | 0.425 | 0.719 | 0.281 |
| 1-3-4 | 0.002 | 0.130 | 0.216 | 0.337 | 0.429 | 0.456 | 0.399 | 0.195 | 0.714 | 0.286 | 0.709 | 0.291 |
| 1-3-5 | 0.082 | 0.435 | 0.249 | 0.418 | 0.487 | 0.542 | 0.550 | 0.378 | 0.609 | 0.391 | 0.725 | 0.275 |
| 1-4-5 | 0.110 | 0.478 | 0.222 | 0.404 | 0.486 | 0.539 | 0.529 | 0.268 | 0.399 | 0.601 | 0.519 | 0.481 |
| 2-3-4 | 0.246 | 0.579 | 0.202 | 0.393 | 0.483 | 0.532 | 0.537 | 0.317 | 0.453 | 0.547 | 0.516 | 0.487 |
| 2-3-5 | 0.170 | 0.577 | 0.203 | 0.396 | 0.481 | 0.537 | 0.529 | 0.293 | 0.591 | 0.409 | 0.725 | 0.275 |
| 2-4-5 | 0.242 | 0.577 | 0.316 | 0.396 | 0.481 | 0.537 | 0.510 | 0.286 | 0.591 | 0.409 | 0.725 | 0.275 |
| 3-4-5 | 0.074 | 0.394 | 0.218 | 0.399 | 0.482 | 0.532 | 0.508 | 0.250 | 0.417 | 0.583 | 0.456 | 0.544 |
| 1-2-3-4 | 0.224 | 0.594 | 0.207 | 0.397 | 0.488 | 0.539 | 0.516 | 0.366 | 0.473 | 0.527 | 0.531 | 0.469 |
| 1-2-3-5 | 0.254 | 0.593 | 0.205 | 0.404 | 0.485 | 0.541 | 0.534 | 0.317 | 0.589 | 0.411 | 0.744 | 0.256 |
| 1-2-4-5 | 0.228 | 0.585 | 0.203 | 0.391 | 0.486 | 0.537 | 0.542 | 0.354 | 0.455 | 0.545 | 0.547 | 0.453 |
| 1-3-4-5 | 0.094 | 0.465 | 0.224 | 0.402 | 0.487 | 0.540 | 0.529 | 0.250 | 0.419 | 0.581 | 0.509 | 0.491 |
| 2-3-4-5 | 0.214 | 0.590 | 0.166 | 0.387 | 0.482 | 0.530 | 0.529 | 0.280 | 0.439 | 0.561 | 0.528 | 0.472 |
| 1-2-3-4-5 | 0.208 | 0.586 | 0.187 | 0.390 | 0.486 | 0.538 | 0.529 | 0.311 | 0.450 | 0.550 | 0.541 | 0.459 |

*Table 19.* Full downstream results for LLaMA 3.1 8B with TIES merging across five domain experts.

| Domain / Stat | Number of experts $k$ | | | | | | | |
|---|---|---|---|---|---|---|---|---|
| | 2 | 4 | 6 | 8 | 10 | 12 | 14 | 16 |
| **Code** | | | | | | | | |
| CE | 0.501872213 | 0.499083328 | 0.493360451 | 0.490470956 | 0.485253937 | 0.4851 | 0.48658 | 0.4914 |
| Var | 7.68E-05 | 0.000147146 | 0.000289833 | 0.000453827 | 0.000232502 | 0.00017 | 0 | |
| Std | 0.008763931 | 0.012130371 | 0.017024468 | 0.021303225 | 0.01524803 | 0.0133 | 0.0063 | |
| **Biology** | | | | | | | | |
| CE | 1.254403344 | 1.187543099 | 1.149616918 | 1.122063748 | 1.091473348 | 1.0853 | 1.06932 | 1.0607 |
| Var | 0.003412081 | 0.006524263 | 0.007148748 | 0.006041633 | 0.004593592 | 0.0027 | 0.000714 | |
| Std | 0.058413017 | 0.080772909 | 0.08455027 | 0.077727944 | 0.067776041 | 0.0521 | 0.02673 | |
| **Physics** | | | | | | | | |
| CE | 1.104021163 | 1.040059072 | 1.006337177 | 0.982421894 | 0.956452194 | 0.9501 | 0.93743 | 0.9293 |
| Var | 0.002224675 | 0.004339474 | 0.004616592 | 0.003374005 | 0.00256163 | 0.0014 | 0.0003 | |
| Std | 0.047166458 | 0.065874686 | 0.067945509 | 0.058086183 | 0.050612551 | 0.0378 | 0.0197 | |
| **Chemistry** | | | | | | | | |
| CE | 1.065878806 | 1.011048543 | 0.981000202 | 0.958958325 | 0.932410791 | 0.9276 | 0.91455 | 0.9067 |
| Var | 0.001910628 | 0.004432202 | 0.004765348 | 0.003875651 | 0.00334207 | 0.0019 | 0.0005 | |
| Std | 0.043710732 | 0.066574781 | 0.069031498 | 0.062254723 | 0.05781064 | 0.044 | 0.0232 | |
| **Geometry** | | | | | | | | |
| CE | 0.499684074 | 0.463583462 | 0.436880708 | 0.422598944 | 0.409463482 | 0.3991 | 0.3922 | 0.3839 |
| Var | 0.000569954 | 0.001058108 | 0.001218494 | 0.001190094 | 0.00067622 | 0.0005 | 0.000259 | |
| Std | 0.023873707 | 0.032528574 | 0.034906937 | 0.034497743 | 0.026004223 | 0.0238 | 0.01609 | |
| **Analysis** | | | | | | | | |
| CE | 0.420332789 | 0.390687826 | 0.368092274 | 0.3561941 | 0.34518962 | 0.3362 | 0.3312 | 0.3247 |
| Var | 0.000428451 | 0.000731257 | 0.000854753 | 0.000915523 | 0.000574783 | 0.00044 | 0.000207 | |
| Std | 0.020699068 | 0.027041758 | 0.029236164 | 0.030257614 | 0.023974637 | 0.0211 | 0.01439 | |
| **Number theory** | | | | | | | | |
| CE | 0.538458845 | 0.502845037 | 0.476979184 | 0.462241288 | 0.447491769 | 0.4345 | 0.42617 | 0.4182 |
| Var | 0.000612441 | 0.00113994 | 0.001340721 | 0.0015108 | 0.000863911 | 0.00067 | 0.00029 | |
| Std | 0.024747554 | 0.033762994 | 0.036615854 | 0.038869017 | 0.029392358 | 0.026 | 0.01723 | |
| **Discrete** | | | | | | | | |
| CE | 0.694258124 | 0.652957155 | 0.62294102 | 0.607414432 | 0.591427917 | 0.5777 | 0.569 | 0.5592 |
| Var | 0.000846054 | 0.001480839 | 0.001805925 | 0.001803155 | 0.0011049 | 0.00086 | 0.0004 | |
| Std | 0.029087002 | 0.038481669 | 0.042496179 | 0.042463568 | 0.033240039 | 0.0293 | 0.0202 | |
| **Algebra** | | | | | | | | |
| CE | 0.419097721 | 0.386756245 | 0.362299539 | 0.349713671 | 0.337287239 | 0.3268 | 0.3204 | 0.3130 |
| Var | 0.000505634 | 0.000875391 | 0.001036457 | 0.001059614 | 0.000619406 | 0.0005 | 0.00023 | |
| Std | 0.0224863 | 0.029587007 | 0.032194052 | 0.032551707 | 0.024887874 | 0.2271 | 0.01537 | |
| **Overall (16-domain macro-average)** | | | | | | | | |
| CE | 0.7774 | 0.7331 | 0.7051 | 0.6874 | 0.6685 | 0.6603 | 0.6509 | 0.6437 |
| Var | 0.0009 | 0.0017 | 0.0021 | 0.0018 | 0.0012 | 0.0006 | 0.0024 | |
| Std | 0.0310 | 0.0418 | 0.0461 | 0.0424 | 0.0357 | 0.0251 | 0.0156 | |

*Table 20.* Cross-entropy (CE), variance, and standard deviation for LLaMA-3.x 3B Instruct in the 16-domain setting (original 9 domains plus Japanese, medical, house-arrangement, Korean, emotion, elementary school mathematics, and Java code). For each domain and each number of merged experts $k \in \{2, 4, 6, 8, 10, 12, 14, 16\}$, we report the mean CE across random expert subsets, along with empirical variance and standard deviation. The overall row represents the macro-average across all 16 domains.

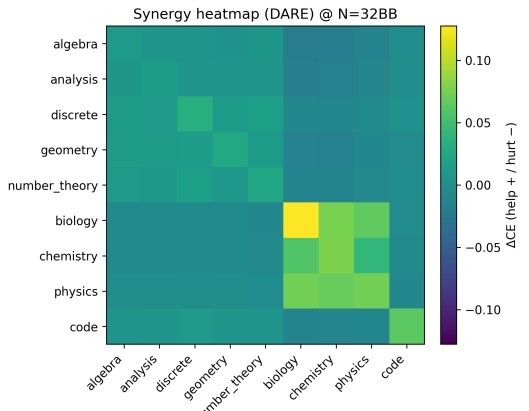
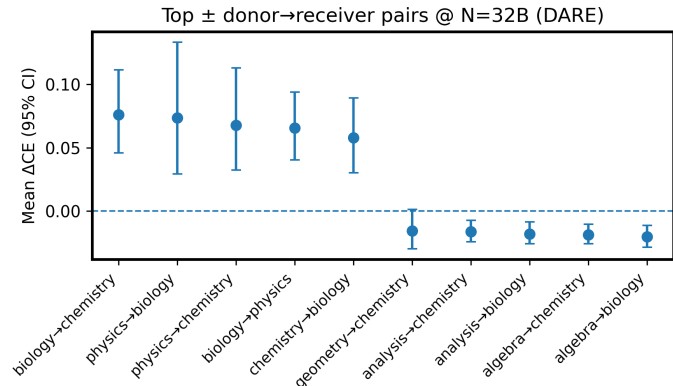

*Figure 21.* Cross-domain synergy (DARE, 32B). Left: synergy heatmap $S_{d \to e}$ (red = help, blue = hurt) showing science↔science and math↔math blocks; cross-block entries are weakly negative; *code*→(*discrete, geometry*) is mildly positive. Right: representative top ± pairs (donor→receiver) highlight actionable donor choices for target domains.

*Table 21.* Across-order dispersion of `Avg.` CE at $k$=1 vs. 8 (DARE). Order sensitivity drops rapidly with $k$ at all $N$ (std $\sim -79\% - 81\%$, range $\sim -83\%$).

| $N$ (B) | $k$ | mean CE | std (across orders) | range (max–min) | CV |
|---|---|---|---|---|---|
| 0.5 | 1 | 0.8164 | 0.0388 | 0.1122 | 0.048 |
| 0.5 | 8 | 0.7810 | 0.0081 | 0.0185 | 0.011 |
| 32 | 1 | 0.5207 | 0.0313 | 0.0865 | 0.060 |
| 32 | 8 | 0.4634 | 0.0060 | 0.0148 | 0.013 |
| 72 | 1 | 0.4638 | 0.0364 | 0.1056 | 0.072 |
| 72 | 8 | 0.4247 | 0.0076 | 0.0179 | 0.018 |

*Table 22.* $\mathrm{Std}_{\mathrm{order}}(N, k) \approx c_0 + c_1/(k+b)$ fits (DARE). A small shared offset $b \approx 2$ with $(c_0, c_1)$ per size explains the decay; $c_0$ is near zero (floor) and $c_1$ shrinks up to mid-scale.

| $N$ (B) | $\hat{b}$ | $c_0$ | $c_1$ | $R^2$ |
|---|---|---|---|---|
| 0.5 | 2.00 | −0.002 | 0.033 | 0.94 |
| 1.5 | 2.00 | +0.002 | 0.028 | 0.90 |
| 3 | 2.00 | +0.003 | 0.023 | 0.88 |
| 7 | 2.00 | +0.002 | 0.021 | 0.92 |
| 14 | 2.00 | +0.003 | 0.019 | 0.91 |
| 32 | 2.00 | +0.001 | 0.017 | 0.75 |
| 72 | 2.00 | +0.002 | 0.023 | 0.69 |

### L.1. Fitted Scaling-Law Parameters on LLaMA Backbones

## M. Cross-Domain Synergy

We quantify *donor–receiver* interactions by adding one expert at a time in the cross-domain setting (randomized orders) and recording the marginal change in macro CE for each evaluation domain, aggregating into a $9 \times 9$ *synergy matrix* $S_{d \to e}$. Using **DARE** at $N$=32B as a representative case (Fig. 21), the heatmap reveals a structured, non-random pattern: science↔science pairs (physics, biology, chemistry) are strongly positive, math↔math pairs are moderately positive, and cross-block interactions are weakly negative at scale; *code* provides mild benefits to *discrete* and *geometry*. This structure is consistent with feature/skill overlap—closer domains supply complementary signal, while distant domains may dilute it—and persists across base sizes with slightly stronger block contrast for larger $N$. In practice, to help a science target, prioritize donors from {physics, biology, chemistry}; for math targets, stay within the math block or include *code*. We report the full matrix values, rank-ordered donor→receiver pairs, and size-wise trends in the released tables and replicate the

*Table 23.* Fitted floor+tail parameters on LLaMA backbones (appendix). Least-squares fits to macro-averaged CE vs. $k$; both series achieve near-unity $R^2$ with a shared $1/(k+b)$ tail.

| Backbone | $R^2$ | $b$ | $L_\infty$ | $A$ | $L(k{=}1)$ | $L(k{=}9)$ |
|---|---|---|---|---|---|---|
| LLaMA-3.2 3B | 0.9989 | 0.6875 | 0.7137 | 0.0783 | 0.7599 | 0.7221 |
| LLaMA-3 8B | 0.9955 | 0.0000 | 0.7252 | 0.0573 | 0.7837 | 0.7325 |

*Table 24.* Fitted scaling-law parameters on LLaMA backbones.

| Backbone | $b$ | $k_{80}$ | $k_{90}$ | $R^2$ |
|---|---|---|---|---|
| LLaMA-3.2 3B | $\approx 1.1$ | $\approx 4$ | $\approx 6$ | 0.999 |
| LLaMA-3 8B | $\approx 1.3$ | $\approx 5$ | $\approx 7$ | 0.995 |

qualitative structure for other methods (TA, TIES) with minor early-$k$ differences that narrow as $k$ grows.

### M.1. Details under DARE

We compute a $9{\times}9$ synergy matrix $S_{d\to e}$ by parsing each DARE trajectory: at step $t$ (sequence model length $t$), adding donor $d_t$ yields a marginal gain $\Delta L_e^{(t)} = L_e^{(t-1)} - L_e^{(t)}$ on evaluation domain $e$, and $S_{d\to e}$ averages these deltas over all occurrences (typically 11~13 per pair at 32B). Using domain blocks $\mathcal{M}=\{$algebra, analysis, discrete, geometry, number_theory$\}$ and $\mathcal{S}=\{$biology, chemistry, physics$\}$, block means $\overline{S}_{A\to B}=\frac{1}{|A||B|}\sum_{d\in A, e\in B} S_{d\to e}$ are: at 7B, $\overline{S}_{\mathcal{M}\to\mathcal{M}}=0.009$, $\overline{S}_{\mathcal{S}\to\mathcal{S}}=0.117$, $\overline{S}_{\mathcal{M}\to\mathcal{S}}=0.014$, $\overline{S}_{\mathcal{S}\to\mathcal{M}}=-0.003$; at 14B, $0.016, 0.077, -0.011, -0.005$; at 32B, $0.012, 0.073, -0.013, -0.005$. The strongest off-diagonal positive pairs at 32B are biology→chemistry ($+0.076$), physics→biology ($+0.074$), physics→chemistry ($+0.068$), chemistry→biology ($+0.066$), biology→physics ($+0.054$); the largest negatives are algebra→physics ($-0.026$), geometry→chemistry ($-0.020$), discrete→chemistry ($-0.018$), algebra→biology ($-0.016$), number_theory→biology ($-0.015$). Donor strengths (row-sums, off-diagonal) rank physics $0.124 >$ biology $0.107 >$ chemistry $0.063 >$ discrete $0.025 \gtrsim$ number_theory $0.021$, with algebra and geometry weakest ($-0.032, -0.005$); receiver susceptibilities (column-sums) rank biology $0.133 >$ chemistry $0.087 >$ physics $0.059$, while code is slightly negative ($-0.029$). Fig. 21 visualizes the 32B heatmap and top pairs that these numbers summarize.

### M.2. Details for Order sensitivity and $1/(k+b)$ fit

From each DARE CSV we derive $k$ (hyphen count $+1$) and collect macro `Avg.` CE across all permutations to compute, per $(N, k)$, the across-order std, range, and CV; we then fit $\mathrm{Std}_{\mathrm{order}}(N, k) = c_0(N) + \frac{c_1(N)}{k+b}$ by grid-search over a small $b\in[0, 2]$ with linear least squares for $(c_0, c_1)$. Table 21 shows that dispersion collapses from $k{=}1$ to $k{=}8$ at 0.5B/32B/72B (std drops $\sim 79\%{-}81\%$, range $\sim 83\%$), and Table 22 reports fitted $(\hat{b}, c_0, c_1)$ and $R^2$ across sizes, where a single small offset $\hat{b}\approx 2$ with $c_0\approx 0$ explains most of the decay; these are the statistics underlying the violin/heatmap/bar visualizations in Fig. 8.

### M.3. Details for Cross-backbone/open-source replication

For each backbone, every CSV row corresponds to one merge order (tokenized in the `model` field) evaluated on a domain with CE in `CE Loss`. We compute macro CE per order by averaging `CE Loss` over the nine domains, derive $k$ as the length of the `model` token list, and then average across orders with the same $k$ to obtain a per-backbone series $\{(k, \bar{L}_k)\}_{k=1}^9$. We fit $L(k) = L_\infty + \frac{A}{k+b}$ by least squares with a small grid over $b \in [0, 1]$; the best $b$ and $(L_\infty, A)$, along with $R^2$ and the end-point values $L(k{=}1)/L(k{=}9)$, are reported in Table 23. These numbers back Fig. 9 and show near-unity $R^2$ and small residuals, confirming that the same $1/(k+b)$ tail holds on LLaMA backbones.

### M.4. Additional Model-Size Slices for Method Comparison

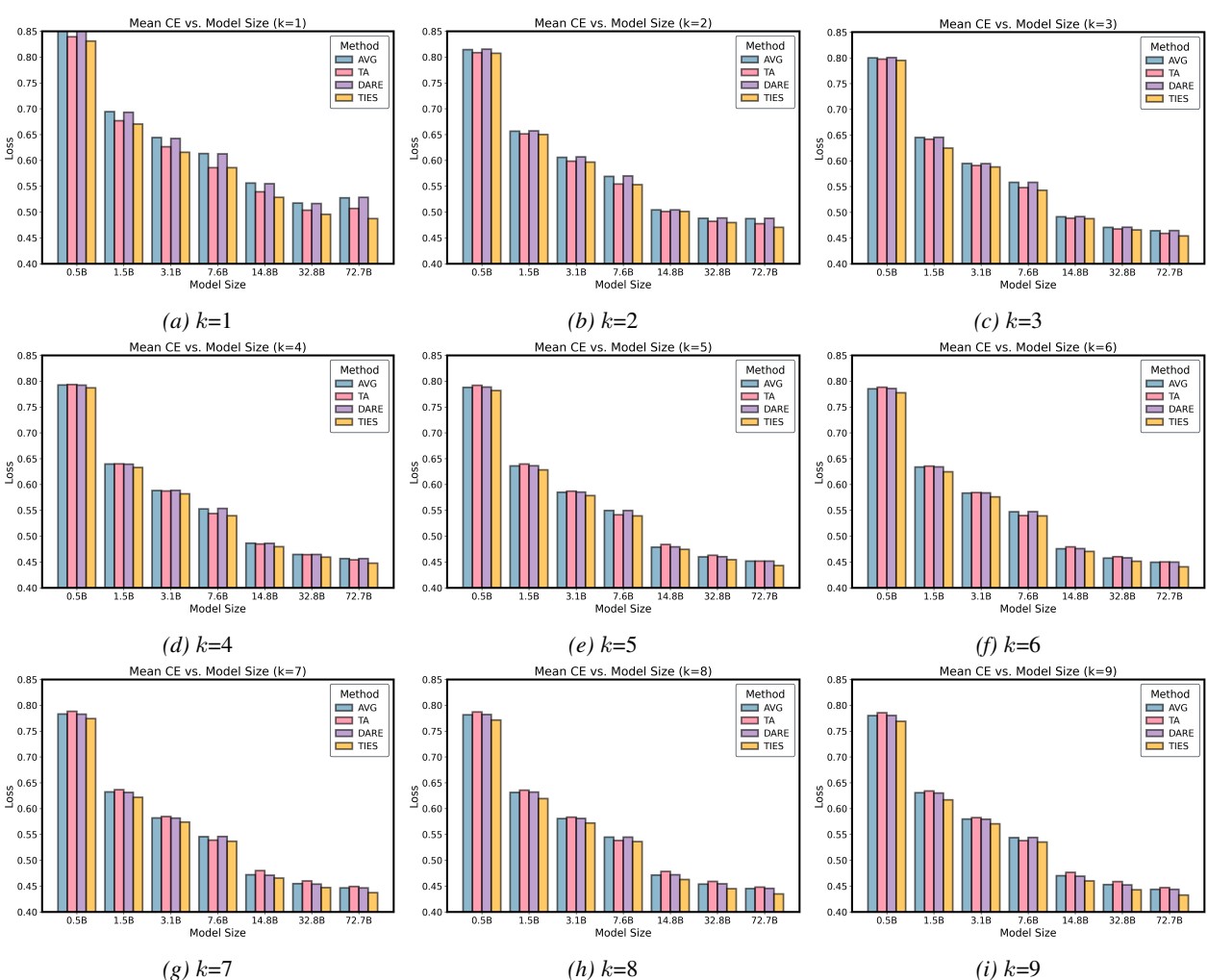

*Figure 22.* Mean CE Loss vs. Model Size with Different $k$

