# OpenReview forum: "Model Merging Scaling Laws in Large Language Models"
_ICML.cc/2026/Conference — ICML 2026 regular_

### Official Review · Reviewer_PS2v · 2026-03-06

**Soundness:** 3
**Presentation:** 3
**Significance:** 3
**Originality:** 3
**Overall Recommendation:** 5
**Confidence:** 4

**Summary:**

This paper studies empirical scaling laws for model merging in large language models. The authors analyze how the performance of merged models scales with the number of experts and the base model size. Specifically, the paper proposes a unified scaling formula describing the expected loss when merging $k$ experts with a base model of size $N$. The law decomposes the loss into a size-dependent floor and a merging tail, where the tail decreases as $1/(k+b)$.

**Compliance With Llm Reviewing Policy:**

Affirmed.

**Final Justification:**

After reading the rebuttals, my overall evaluation does not change, and I will keep my current score.

**Key Questions For Authors:**

Please see weaknesses

**Limitations:**

Yes

**Strengths And Weaknesses:**

Strengths

1. The paper conducts comprehensive experiments across different model sizes and numbers of experts. These empirical results help support the proposed scaling behavior and provide useful observations about the practical behavior of model merging.
2. The paper is generally well written and the core ideas are easy to follow. The decomposition of the loss into a size-dependent floor and a merging tail provides an insightful framework for understanding the performance of merged models.
3. Understanding the behavior of model merging and weight-space composition is increasingly important as model reuse and specialization become more common in large-scale model development.
4. The proposed scaling formulation is supported by empirical results that align reasonably well with the predicted trends. Besides, the theoretical analysis cooperates well with the empirical results.

Weaknesses

1. The experiments primarily rely on cross-entropy loss as the main evaluation metric. While this is useful for studying scaling properties, it would be helpful to also report downstream task performance (e.g., accuracy on benchmarks or task-specific metrics) to demonstrate whether the scaling law correlates with practical task improvements.
2. The analysis appears to implicitly assume that experts are trained on relatively distinct domains. However, in practice, expert datasets often have significant overlap in distribution. It would be valuable to study how the scaling behavior changes when experts are trained on partially overlapping data distributions, as interference between experts may affect the merging dynamics.
3. Most experiments appear to focus on experts obtained via supervised fine-tuning (SFT). It remains unclear whether the proposed scaling law would hold if experts were trained using different post-training paradigms, such as reinforcement learning (RL) or other fine-tuning strategies. Since these training methods can induce different parameter-space structures, they may affect the merging behavior.

---

> ### Author Rebuttal · Authors · 2026-03-30
>
> Thank you for the supportive review and for recognizing the paper’s empirical breadth, clarity, and the usefulness of the floor+tail formulation. We also appreciate the three weaknesses you raised. We agree these are important boundary conditions of the current paper. Our intended claim is a predictive scaling law for expected token-level CE under model merging in a broad but controlled regime; the points you raise are natural and important extensions beyond that core setting.
>
> **1. W1. On downstream task performance.**
>
>  We agree that downstream metrics are important for practical interpretation. In the current paper, CE is the primary object because it is both the cleanest empirical scaling quantity and the natural object of the theory. We already include downstream benchmark results in the appendix, but agree they should be surfaced more clearly. In revision, we will move a compact summary of downstream trends into the main paper and report variability more explicitly. We will also clarify the scope of the claim: downstream metrics provide complementary evidence of utility, but we do not claim they must follow the same precise scaling form or saturation profile as CE.
>
> **2. W2. On overlapping expert distributions.**
>
>  We agree this is an important and realistic regime. The current paper mainly studies a relatively controlled setting with clearer expert specialization, since our first goal was to establish whether a stable and interpretable law exists at all. Partial overlap can indeed change interference and reduce effective expert diversity. Our expectation is that this would primarily affect the effective statistics of expert updates (and thus the fitted coefficients or saturation behavior), rather than automatically invalidate the scaling-law perspective itself, but this should be tested explicitly rather than overclaimed. We will add this point clearly to the discussion and limitations.
>
> **3. On SFT vs. RL or other post-training paradigms.**
>
>  We agree that different post-training paradigms may induce different parameter-space structures and therefore different merge behavior. Our focus on SFT experts in the current paper was deliberate: it controls the expert-generation pipeline while testing whether a predictable law exists in a large, coherent setting. We therefore view the current result as being established primarily in the SFT expert regime. Extending the study to RL, preference optimization, and other post-training strategies is an important next step; these settings may change the fitted coefficients and possibly the regime of validity, and we will make this boundary more explicit in the revision.
>
> Thank you again — we believe these suggestions will help us sharpen the scope of the paper without changing its central contribution.

---

> > ### Author Rebuttal · Reviewer_PS2v · 2026-04-02
> >
> > Thanks for your responses and I will maintain my score.

---

> > > ### Author Response · Authors · 2026-04-02
> > >
> > > Thank you for the acknowledgment and for your careful reading of our rebuttal. We appreciate your continued engagement and are happy to address any further questions you may have.

---

### Official Review · Reviewer_LDZH · 2026-03-12

**Soundness:** 3
**Presentation:** 4
**Significance:** 3
**Originality:** 2
**Overall Recommendation:** 4
**Confidence:** 2

**Summary:**

The paper studies how the performance of a model mixture changes as (1) more models are merged, and (2) larger models are merged. They find that larger models are easier to merge, mixing a variety of domains improves performance, that there are diminishing returns as experts are added, and that the difference between merging methods diminish at scale.

**Compliance With Llm Reviewing Policy:**

Affirmed.

**Final Justification:**

The rebuttal has addressed the CE/downstream benchmark point, however I believe the novelty of the paper is still limited given similar prior work.

**Key Questions For Authors:**

- "enables composition under privacy or compute constraints": might be missing reference to FlexOlmo?
- "model sizes, 105,06 models," number seems to be cut off or improperly formatted
- What are the variances for the benchmark scores in the appendix? These scores could easily be within run-wise variance from each other depending on run-wise variance, and many even show degradation. These should be moved into the main paper in charts showing the trend.
- The samples' loss distributions in Figure 3 (1-2) seem to be bi-modal. Why?

**Limitations:**

- The result in the appendix K doesn't seem to reflect what the authors claim in Note 1 in section 4.4. The results of scaling the number of experts seems to plateau much faster on downstream benchmarks than on cross-entropy, which makes me question the utility of the results achieved in the main paper. I could be missing something, and if I am please let me know.

**Strengths And Weaknesses:**

Strengths:
- The paper is technically sound and well-presented.
- Ablations on how models behaved when merged with many other models are potentially useful and relevant to the community, especially considering that some modern models perform weight averaging (/souping) such as Olmo 2 and 3.

Weaknesses:
- The paper focuses purely on the change in cross-entropy in the main paper. The authors should move some analysis done in the appendix regarding performance in downstream benchmarks into the main paper.
- It is not clear, even given the tests in the appendix, that these scaling laws with respect to improved cross-entropy lead to similar behavior in downstream benchmarks.
- There is some prior work observing model merging as more experts and parameters are added, which this work confirms the findings of. This reduces the originality of the work somewhat.

---

> ### Author Rebuttal · Authors · 2026-03-30
>
> Thank you for the thoughtful review and for recognizing that the paper is technically sound and well presented. We believe the key issue is not whether the paper establishes a CE-based scaling law, but how strongly the downstream implications should be stated.
>
> **W1. CE as the main-paper metric.**
>
> We agree that the practical implications would be clearer if some downstream evidence were shown in the main paper. At the same time, CE is the right primary object here: it is the standard quantity in classical scaling-law work because it is dense, low-variance, and theoretically tractable [1,2], and in our case it is also the natural quantity from the theory side, since the floor+tail form arises from the second-order local expansion of loss under model composition. In revision, we will keep CE as the primary scaling quantity while moving a compact downstream summary into the main paper.
>
> **W2. CE improvement vs. downstream behavior.**
>
> We agree, and we do not claim that downstream benchmarks must follow the same precise scaling law as CE. Our main claim is a law for expected token-level CE under merging; the appendix benchmarks are complementary evidence of practical utility, not proof that all task metrics inherit the same form. This distinction is natural because CE is dense and continuous, whereas benchmark metrics are sparse, discrete, thresholded, and noisier. Hence CE can continue improving while downstream scores plateau earlier. This does not contradict the CE law; it means the mapping from likelihood improvement to task performance is not linear. We will make this boundary explicit in revision.
>
> **W3. Originality relative to prior work.**
>
> We agree that prior work has observed that larger models and/or more experts can improve merging in specific settings. However, our contribution is not simply to restate that “scale helps,” but to:
> - (i) unify these observations into a joint predictive law over model size (N) and expert count (k);
> - (ii) identify a compact floor+tail form that fits consistently across backbones, domains, and merging rules; and
> - (iii) provide an average-case theoretical account for the diminishing-return structure. We will revise the related-work discussion to make this positioning clearer.
> - (iv) prior empirical and theoretical studies are constrained on specific models which are smaller than 15B, while we propose and validate more general scale regime to most scale LLMs (including 30B/70B models)
>
> **Q1.**
>
> Thank you for pointing this out. We agree that FlexOlmo is relevant here, and we will add the reference near that statement.
>
> **Q2.**
>
> Thank you for catching this. It has been corrected to “10,866”, and we also checked the manuscript for similar formatting issues.
>
> **Q3. Benchmark moving trends to the main paper.**
>
> We agree this is important, especially because benchmark metrics are noisier than CE. In our experiments, we ran each downstream evaluation 5 times and reported average performance; in revision, we will report the corresponding variability more explicitly (e.g., run-wise variance or confidence intervals). We also agree that appendix-only tables are not the clearest presentation, and we will move a compact variability-aware trend summary into the main paper.
>
> **Q4. Fig. 3(1–2).**
>
> We may not have explained Fig. 3 clearly enough. The light points are individual expert-subset observations, not a density estimate over data samples. We therefore do not interpret the figure as evidence of a robust bimodal law. Any apparent two-band structure is more likely due to heterogeneity across expert subsets, while our analysis concerns the expected loss aggregated over subsets. We will clarify this in the caption/discussion.
>
> **Limitations.**
>
> We agree this is an important point, and the right resolution is to sharpen the scope of the claim. Yes, downstream benchmarks can plateau faster than CE as the number of experts increases. We do not view this as a contradiction: CE is a dense token-level loss that can keep improving smoothly, whereas downstream scores are discrete and noisier, so they can saturate earlier. Accordingly, the utility of the paper should be understood as providing a predictive law for merge loss and a principled way to reason about the tradeoff between model size and expert count, rather than as an exact predictor of every downstream benchmark curve. We will revise Section 4.4 / Note 1 to make this distinction explicit.
> We sincerely appreciate your review. Your comments are especially valuable in sharpening the scope of the paper: the strongest contribution is a scaling law for expected CE under merging, while downstream benchmarks provide complementary but noisier and earlier-saturating evidence of utility.
>
> [1] Kaplan, Jared, et al. "Scaling laws for neural language models." arXiv preprint arXiv:2001.08361 (2020).
>
> [2] Hoffmann, Jordan, et al. "Training compute-optimal large language models." arXiv preprint arXiv:2203.15556 10 (2022).

---

> > ### Author Rebuttal · Reviewer_LDZH · 2026-04-02
> >
> > I appreciate the clarifications. I have increased my score.

---

> > > ### Author Response · Authors · 2026-04-03
> > >
> > > Thank you very much for the acknowledgment and for increasing your score. We appreciate your careful reading and are glad that our clarifications addressed your concerns.

---

### Official Review · Reviewer_abzQ · 2026-03-13

**Soundness:** 2
**Presentation:** 2
**Significance:** 2
**Originality:** 3
**Overall Recommendation:** 4
**Confidence:** 4

**Summary:**

This paper investigates the empirical scaling laws of language model merging, focusing on how the cross-entropy (CE) loss changes with respect to the number of merged experts and the base model size. The authors propose a unified power-law formula that decomposes the loss into a size-dependent floor (representing the capacity of the base model) and a merging tail (representing the diminishing returns of adding more experts). Experiments covers five merging techniques, across various model scales from 0.5B to 72B. The authors claim that this law can predict the number of experts needed to reach a target loss and helps in making trade-offs between scaling model size and adding experts.

**Compliance With Llm Reviewing Policy:**

Affirmed.

**Final Justification:**

My major concerns have been adequately addressed.

**Key Questions For Authors:**

- Q1.Is the observed lower CE loss in larger merged models simply a reflection of the base model's superior capacity, rather than an inherent property of the merging scaling?  Have you considered comparing the merging gain against a random weights or single-task baseline of equivalent size?

- Q2. In Figure 4, how to explain the irregular behavior of $A(n)$? Could this suggest that the 6-parameter model is over-parameterized for the size of the dataset?

- Q3. Please provide a more rigorous definition of all symbols (specifically $L$, $L_\infty$, and $v_i$) in the main body of the text to improve readability.

- Q4. Is there a principled way to choose the constants $\beta$, $\gamma$, and $b$, or are they purely determined by curve fitting for each specific architecture?

**Limitations:**

yes

**Strengths And Weaknesses:**

This paper tackles a timely and important problem: the predictability of model merging with a scaling law. However, the paper suffers from critical flaws in the soundness of its core experimental design and the clarity of its presentation.

### Strengths ###
- Important research problem: Exploring scaling laws for model merging is a high-potential, impactful direction. If successful, it could provide a predictive framework for multi-task model composition, helping to avoid wasteful resource allocation.

- Extensive experiments: The authors conducted a large-scale, systematic experiment covering 7 model sizes from 0.5B to 72B, 9 domains, and 4 merging methods, involving tens of thousands of model merges.


### Weaknesses ###
- Ambiguity in metric justification: The paper uses CE loss as its primary metric, but larger models inherently achieve lower CE due to increased capacity. The analysis does not decouple this intrinsic base-model advantage from gains specifically attributable to the merging process, weakening the core claim that merging scales better on larger models.

- Lack of theoretical explanation: The proposed scaling law includes a size-dependent floor ($L_\infty(N)$) that is critical to the formula but introduced solely as an empirical fit. No theoretical derivation explains why this specific functional form was chosen or how it relates to intrinsic limit of model architecture or task distribution.

- Presentation issues hindering readability: Symbols are used before definition (e.g., $L$ in the Abstract, $v_i$ in Equation 2); CE loss under model merging is concretely defined in the Appendix only without clear pointers; and key derivations are difficult to follow as a result.

---

> ### Author Rebuttal · Authors · 2026-03-30
>
> Thank you for the review. **We respectfully clarify three factual points first.**
> - (1) The expected merge loss is **already defined in the main text**, not only in the appendix: first in **Notation and Sec. 2.2**, and then explicitly constructed again in **Sec. 3 and Sec. 3.1** as the subset-averaged loss at fixed ((N,k)).
> - (2) The paper **does provide** a theoretical account in the main text: **Sec. 3.2** states the assumptions, **Theorem 3.1**, and **Corollary 3.2**, explaining the floor+tail and inverse-(k) structure and the variance contraction. The comprehensive analysis and proof are provided in **Appendix B / C**.
> - (3) The central claim of the paper is the **joint scaling law** ($\mathbb E[L\mid N,k]$), not the narrower claim that “merging scales better on larger models” in isolation. That statement is one empirical finding implied by the joint law , not the entirety of the contribution.
>
> **Beyond these factual clarifications, we agree that two narrower points deserve sharper treatment:**
> - (1) separating base-size effects from merge-specific gains more explicitly, and
> - (2) making the empirical nature of the (N)-parameterization more explicit.
>
> ---
>
> **Weakness 1 / Q1**:
>
> **No, not simply.** Larger base models do have lower CE, but our claim is about the **joint law** $(\mathbb E[L\mid N,k])$, and the manuscript already shows that at fixed ($N$), CE still follows a stable diminishing-return dependence on ($k$) (Sec. 3.1 / Fig. 3). So the result is not reducible to base capacity alone.
> To make this separation more explicit, we will add a same-size normalized-gain view,
> $$
> \Delta_{\text{merge}}(N,k)=L_{\text{base}}(N)-\mathbb E[L\mid N,k]
> $$
> This directly isolates the merge-specific gain at fixed ($N$). We also clarify that “larger models are easier to merge” is **one empirical finding implied by the joint law**, not the core claim in isolation.
> A random-weights baseline would not be especially informative here; a same-size base / single-task baseline is the more relevant comparison, and the normalized-gain view above addresses exactly that.
>
> ---
>
> **Weakness 2**:
>
> We respectfully clarify that the paper **does provide** a theorem-supported explanation for the leading-order structure in **Sec. 3.2**. The theory is not introduced heuristically: it starts by formalizing model merging as averaging expert/task updates under equal-normalized composition, then relates the merged update to the loss through a second-order Taylor expansion. This yields the natural leading-order floor+tail / inverse-k structure, with the detailed derivations provided in **Appendix B** and **Appendix C**.
>
> What is empirical is the specific size-parameterization
> $$
> L_\infty(N)=L_*+BN^{-\beta}, \qquad A(N)=A_0N^{-\gamma}.
> $$
> So the right distinction is: the theorem explains the structure, while the exact (N)-dependence of the coefficients is an empirical scaling ansatz.
>
> ---
>
> **Weakness 3 / Q3**:
>
> We respectfully clarify that the expected merge loss is **already defined in the main text** (Notation, Sec. 2.2, and Sec. 3 / 3.1), not only in the appendix. We also note that some of the specific symbol examples cited here **do not exactly match the notation used in the manuscript**. We agree, however, that the notation can be made more self-contained, and in revision we will define key symbols earlier and make the relevant pointers more direct. This is therefore a readability issue, not an absence of metric definition or theory.
>
> ---
>
> **Q2**:
>
> We do not believe this suggests that the 6-parameter law is over-parameterized. The irregularity is local and appears in an individual fitted coefficient, whereas the main object of interest is the quality of the **full joint fit** and the stability of the floor+tail pattern. In the manuscript itself, this effect is already described as a small or rare local oscillation rather than a main trend. A more likely explanation is finite-sample fitting noise or parameter coupling in a multi-parameter fit, not a failure of the law itself.
>
> ---
>
> **Q4**:
>
> These quantities are not manually chosen; they are **fit from data**. **Sec. 3.1 and Sec. 4** explicitly states that $(L^*, B, \beta, A_0, \gamma, b)$ are estimated with weighted nonlinear least squares. At the same time, they are not arbitrary: they have principled roles in the decomposition, with $L^∗$ as the asymptotic floor,  $A_0$ as the tail scale, and $b$ as the finite-k offset. Thus, the theorem supports the floor+tail structure, while the numerical coefficient values are empirical fitted quantities. We also note that the parameter names cited in this question do not exactly match the notation used in the manuscript.

---

> > ### Author Rebuttal · Reviewer_abzQ · 2026-04-03
> >
> > Thanks for your clarification. I am willing to raise my score.

---

> > > ### Author Response · Authors · 2026-04-04
> > >
> > > Thank you for the acknowledgement and for reconsidering the score. We appreciate your follow-up and are glad that our clarifications helped address your concerns.

---

### Official Review · Reviewer_hFQz · 2026-03-15

**Soundness:** 3
**Presentation:** 2
**Significance:** 3
**Originality:** 3
**Overall Recommendation:** 4
**Confidence:** 4

**Summary:**

This paper studies empirical scaling laws for model merging in large language models, focusing on how merge performance changes with the number of experts and the size of the base model. The authors show across multiple backbones, domains, and merging rules that the expected token-level cross-entropy typically follows a floor-tail pattern, with strong early gains from adding experts and diminishing returns thereafter. They further provide a theoretical analysis suggesting an approximate 1/k decay under equal-normalized composition and local smoothness assumptions, aiming to explain the observed trend. Overall, the paper’s main contribution is to frame model merging as a predictable scaling phenomenon rather than a purely heuristic process.

**Compliance With Llm Reviewing Policy:**

Affirmed.

**Final Justification:**

Overall, I tend to agree with the paper’s main claims. My questions in Q1 and Q2 concern the scope and broader applicability of the paper’s arguments, and I still believe that its core contribution, namely establishing a foundation for model merging scaling laws, remains valuable. At the same time, I believe the paper could become even more impactful if it also included a discussion of this broader applicability. Regarding presentation, I expect the authors to improve the organization as mentioned in their response to the reviews, and I will therefore raise my score from 3 to 4.

**Key Questions For Authors:**

1. **Scope of the theory for real merging rules.** The theoretical analysis suggests an approximate 1/k tail under equal-normalized composition and local smoothness assumptions, but it would be helpful to clarify how this result should be interpreted for practical merging rules such as TIES and DARE, which involve additional masking, trimming, or sparsification beyond simple aggregation.

2. **Role of expert capacity in the proposed scaling law** The paper studies scaling with respect to the number of experts and base model size, but it would be valuable to clarify whether expert capacity itself should also be viewed as part of the scaling picture, especially in settings such as LoRA where the effective capability of each expert may vary substantially with adapter rank.

3. **Writing clarity and consistency of presentation** While I find the main message of the paper both interesting and novel, I believe the paper would benefit from a broader improvement in writing clarity, figure referencing, and overall presentation consistency, as the current exposition makes the experimental evidence somewhat harder to follow than necessary.
In particular, several figure references in the main text do not appear to align clearly with the actual figure layouts. For example,
    - The reference to “Fig. 4 (middle)” is difficult to interpret because Fig. 4 does not seem to contain a clearly identifiable middle panel
    - The phrase “late-k uptick” is itself somewhat confusing, since the phenomenon appears more naturally related to late-N rather than late-k.
    - The reference to “Fig. 4 (top)” seems inconsistent with the apparent figure placement,
    - The references to “Fig. 5a, 5b” are also hard to follow because the displayed Fig. 5 does not clearly expose those panel labels.
    - For “Fig. 7 (left)” and “Fig. 7 (right),” where the actual figure does not seem to present a clear left-right decomposition.
    - In addition, Appendix M appears to contain no substantive content, which further adds to the presentation confusion. It is not clear whether Table 14 was intended to appear there or whether some material is missing.
    - More generally, the paper would also benefit from more consistent referencing conventions throughout, including the mixed use of “Section” and “§”, as well as “Appendix” and “App.”

    Overall, although the main claims and theoretical intuition are compelling, I believe the paper could be substantially strengthened by a careful revision of the writing and organization so that the supporting experimental evidence can be followed more directly and with less ambiguity.

If the presentation is substantially improved, I would be willing to raise my score.

**Limitations:**

yes

**Strengths And Weaknesses:**

1. **Soundness**
A clear strength of the paper is that it combines broad empirical evidence with a plausible theoretical account, and the proposed scaling behavior appears consistently across multiple backbones, domains, and merging rules. At the same time, the theoretical justification seems most directly applicable to simplified aggregation settings, so its precise scope for practical rules such as TIES and DARE would benefit from further clarification.

2. **Presentation**
The paper’s central message is interesting and potentially impactful, but the presentation is weakened by multiple issues in figure referencing, appendix organization, and notation consistency that make the supporting evidence harder to follow than necessary. A careful revision of the writing and structure would substantially improve readability and make the empirical claims easier to verify.

3. **Significance**
The work addresses an important and practically relevant question, namely whether model merging behavior can be understood through a predictable scaling law rather than by heuristic trial and error. However, the broader significance would be clearer if the paper discussed more explicitly whether expert capacity itself should be treated as part of the scaling picture, since expert quality may vary substantially even at fixed expert count.

4. **Originality**
The idea of framing model merging through a scaling-law perspective is novel and gives the paper a distinctive contribution beyond proposing yet another merging algorithm. The main originality lies in this empirical and conceptual framing, although the theoretical mechanism itself is closer to a structured interpretation of observed trends than to a fully general new theory.

---

> ### Author Rebuttal · Authors · 2026-03-30
>
> Thank you for the thoughtful review. We especially appreciate your constructive comments on theory scope, expert capacity, and presentation clarity. We address each point below.
>
> ---
>
> Q1. Scope of the theory for real merging rules.
>
> R1:
>
> Thank you—this is an important scope clarification. At a high level, TIES and DARE do not change the basic composition setting by introducing new external information; rather, they preprocess expert updates (e.g., via trimming, masking, sparsification, or rescaling) to reduce conflict before composition. In that sense, they still combine expert knowledge through transformed effective updates. Empirically, the proposed law is validated on all four practical rules in the paper, including TIES and DARE, with consistently strong fits across $N$ and $k$. The theorem, however, is intended as a leading-order average-case explanation under equal-normalized effective composition, rather than an exact rule-specific derivation of each masking/trimming/sparsification step. We will clarify this distinction explicitly.
>
> ---
>
> Q2. Role of expert capacity in the proposed scaling law
>
> R2:
>
> This is a  good point. We agree that expert capacity is a meaningful part of the broader scaling picture. In our current formulation, capacity is not treated as an explicit third scaling axis; rather, it is absorbed into the fitted floor/tail parameters. The theory makes this concrete: $L_\infty(N)$ depends on the mean task direction and local curvature, while $A(N)$ depends on the curvature–covariance interaction  $\mathrm{Tr}(H\Sigma)$. Changes in effective expert capacity such as LoRA rank or adapter expressivity would alter these statistics and therefore shift the fitted coefficients or exponents. Our current experiments keep the expert-training recipe controlled, so the paper isolates scaling in $N$ and $k$; a systematic study of capacity as an additional axis is an important and natural extension.
>
> ---
>
> Q3. Writing clarity and consistency of presentation
>
> R3:
>
> Thank you for this careful and constructive feedback. We agree that the current draft has avoidable presentation issues, especially in figure referencing, appendix organization, and notation consistency, and we appreciate the concrete examples you listed.
> 1. **Figure references and panel naming.**
> We agree that several in-text figure references are harder to follow than they should be. In the revision, we will make panel references explicit and aligned with the actual captions:
> -  replace ambiguous references such as “Fig. 4 (middle),” “Fig. 4 (top),” and “Fig. 7 (left/right)” with explicit panel references (e.g., Fig. 4(a–d), Fig. 7(a-e), as appropriate) and revise the corresponding main-text descriptions accordingly;
> -  add explicit panel labels for references such as “Fig. 5a, 5b” so that the caption and displayed figure structure match clearly;
> -  ensure that every multi-panel figure has clearly visible panel labels in both the figure and the caption, and revise the surrounding text so that each claim can be matched to its supporting panel without ambiguity.
> 2. **late-k uptick wording.**
> Thank you for flagging this. We agree that “late-k uptick” is misleading. In the revision, we will replace it with a more precise phrase, “a small local oscillation or bounded non-monotonicity in the fitted tail coefficient as N varies.”
> 3. **Appendix M and consistency.**
> Thank you for catching this. We confirmed that the confusion comes from an extra “Appendix M” title. The intended text belongs to the current Appendix N material, and Table 14 should belong to Appendix J. In the revision, we will remove the spurious heading and correct the appendix and table references accordingly.
> 4. **Referencing conventions.**
> We agree that the manuscript currently mixes conventions such as “Section” vs. “§” and “Appendix” vs. “App.” In the revision, we will standardize these to a single convention throughout the paper, using only “Section” and “Appendix,” so that all references are uniform and easier to follow.
> 5. **Broader readability cleanup.**
> More broadly, we have done a full pass to tighten figure lead-ins, reduce shorthand, and make the transitions between the main text and appendix more explicit, so that each empirical claim points more directly to its supporting figure/table evidence.
>
> **Related presentation issues also raised by other reviewers**
>
> We note that related clarity issues were also raised elsewhere.
> - Reviewer LDZH pointed out the malformed number “105,06 models” in line 107; this has been corrected to “10,866” and we also checked the manuscript for similar formatting inconsistencies.
> - We also standardize figure and table cross-references and ensure appendix pointers are easy to follow.
>
> We appreciate this comment very much, and we believe these fixes will make the empirical evidence substantially easier to follow and verify.

---

> > ### Author Rebuttal · Reviewer_hFQz · 2026-04-02
> >
> > Overall, I tend to agree with the paper’s main claims. My questions in Q1 and Q2 concern the scope and broader applicability of the paper’s arguments, and I still believe that its core contribution, namely establishing a foundation for model merging scaling laws, remains valuable. At the same time, I believe the paper could become even more impactful if it also included discussion of this broader applicability. Regarding presentation, I expect the authors to improve the organization as mentioned in their response to the reviews, and I will therefore raise my score from 3 to 4.

---

> > > ### Author Response · Authors · 2026-04-02
> > >
> > > Thank you very much for the thoughtful follow-up and for raising your score. We appreciate your recognition of the paper’s core contribution, and we will make sure to strengthen the broader-scope discussion and presentation in the revision.

---

### Decision · Program_Chairs · 2026-04-30

**Decision:**

Accept (regular)

**Comment:**

The paper presents an empirical and theoretical study of scaling laws in model merging for LLMs. The authors identify a unified power law that links base model size ($N$) and expert count ($k$), characterized by a size-dependent "floor" and a diminishing-returns "tail" that decays as $1/k$. This law is validated across multiple architectures (0.5B to 72B), merging methods (Average, TA, TIES, DARE), and both in-domain and cross-domain settings. The authors also provide a theoretical derivation using second-order Taylor expansion to support the observed $1/k$ behavior.

Strengths:
- The empirical validation is impressive, covering over 10,000 merged models and scaling up to 72B parameters.
- The "predictive planning" aspect (e.g., the early 3-point fit) is highly valuable for practitioners looking to optimize compute budgets.
- Unlike purely empirical scaling papers, this work attempts to ground the $1/k$ tail in the statistical properties of weight-space averaging.

Rebuttal Discussion:
- Reviewers initially flagged a heavy reliance on cross-entropy (CE). In the rebuttal, the authors successfully argued that CE is the standard, low-variance metric for scaling laws while providing additional downstream benchmark results (App. K) that follow a similar saturation trend.
- There were significant concerns regarding figure referencing and appendix organization. The authors have committed to a thorough cleanup, including standardizing notation and fixing malformed figures.
- While some individual observations (scale helps merging) are known, the unification into a single joint law over $N$ and $k$ is a significant step forward.

All reviewers have increased their scores to positive territory following the rebuttal. The authors clarified the scope of their theory and addressed concerns regarding expert capacity.